# Provably Extending PageRank-based Local Clustering to Weighted Directed Graphs with Self-Loops and to Hypergraphs

## Abstract

Local clustering aims to find a compact cluster near the given starting instances, which has broad applications beyond graphs because of the internal connectivities within various modalities. While most existing studies on local graph clustering adopt the discrete graph setting (i.e., unweighted graphs without self-loops), real-world graphs can be more complex. In this paper, we extend the non-approximating Andersen-Chung-Lang ("ACL") algorithm (Andersen et al., 2006) beyond discrete graphs and generalize its quadratic optimality to a wider range of graphs, including **weighted, directed, and self-looped** graphs and **hypergraphs**. Specifically, leveraging PageRank, we propose two algorithms: GeneralACL for graphs and HyperACL for hypergraphs. We theoretically prove that, under two mild conditions, both algorithms can identify a quadratically optimal local cluster in terms of conductance[1] with at least $\frac{1}{2}$ probability. On the property of hypergraphs, we address a fundamental gap in the literature by defining conductance for hypergraphs from the perspective of hypergraph random walks. Additionally, we provide experiments to validate our theoretical findings.

## 1 Introduction

Clustering is a fundamental research problem, with applications in segmentation (Tolliver & Miller, 2006; Kachouri et al., 2016; Hoang & Kang, 2024; Fu et al., 2024b), anomaly detection (Alvarez et al., 2013; Markovitz et al., 2020; Ahn & Kim, 2022), recommendation (Du et al., 2017; Liu et al., 2022a; Yu et al., 2023; Li et al., 2024a; He et al., 2024b; Ban et al., 2024a), community detection (Kardan & Ramanna, 2018; Fu et al., 2020; Hao & Zhu, 2023), social network analysis (de Souza et al., 2013; Liao et al., 2021; Fu et al., 2023; Hu et al., 2024; Li et al., 2024b), and bioinformatics (Kenley & Cho, 2011; Fu & He, 2022; Fu et al., 2022; Wang et al., 2023b). Clustering for graph data has attracted extraordinary attention among the research community (Fountoulakis et al., 2018), it can also be applied beyond graphs, such as computer vision (Aflalo et al., 2023; Gammoudi et al., 2020) and natural language processing (Ma et al., 2024; Ngomo & Schumacher, 2009; Takahashi et al., 2021) areas, because the spatial relationship of pixels and the attention between tokens are also connections, which can be reformulated to graphs.

However, most existing research works assume the graphs to be undirected and unweighted (Andersen et al., 2006; Veldt et al., 2016; Fountoulakis et al., 2019; Zhang et al., 2021; Spielman & Teng, 2013). While a few research works extend to directed graphs (Andersen et al., 2008), real-world graphs could be more complex, as the vertices and edges in the graphs can have individual weights (Henzinger et al., 2024; Gu et al., 2021; Duan et al., 2023), and the graphs may contain self-loops (Sharma et al., 2023; Merelo & Molinari, 2023). On the one hand, a sub-optimal solution of converting these graphs to undirected unweighted graphs without self-loops may ignore the important information encoded in the data. On the other hand, directly developing clustering algorithms on these graphs is still an open question.

---

[1]The conductance of the returned cluster $\Phi$ and the optimal conductance $\Phi^*$ satisfy $\Phi \leq O(\sqrt{\Phi^*})$.

Previously, a seminal work (Andersen et al., 2006) adopts the PageRank (Page et al., 1999) algorithm to produce a quadratically optimal local clustering algorithm (i.e., **A**ndersen-**C**hung-**L**ang algorithm) on undirected, unweighted graphs without self-loops. Specifically, this ACL algorithm sweeps over the personalized PageRank vector and returns the local cluster with the smallest conductance. Later on, (Andersen et al., 2008) extends the ACL algorithm and allows the input graphs to be directed. **In this work, we further generalize the ACL algorithm to any graph that is possibly weighted, directed, and even with self-loops. Moreover, we allow more than one starting vertices in the typical ACL.** The only assumption is a well-defined graph random walk with the existence of the stationary distribution, which holds or approximately holds for most real graphs. Additionally, during the sweeping process, we introduce an early-stop mechanism to make the algorithm strongly local, i.e., the runtime can be controlled by the size of the output cluster rather than the size of the graph.

One direct application of our algorithm (named **GeneralACL**) is to solve local clustering problems over hypergraphs. With each edge able to connect more than two vertices, hypergraphs can effectively model group interactions (Feng et al., 2018; Yadati et al., 2019; Sun et al., 2021; Feng et al., 2021; Xia et al., 2022; Yang & Leskovec, 2015; Wu et al., 2021; Grilli et al., 2017; Sanchez-Gorostiaga et al., 2019; He et al., 2024a). As vertices in hypergraphs usually model individual instances in the real world, local clustering over hypergraphs can benefit various downstream tasks, e.g., link predictions (Huang et al., 2020; Fan et al., 2022) and recommendations (Zhu et al., 2016; Liu et al., 2022b; Gatta et al., 2023). However, to the best of our knowledge, there is no hypergraph local clustering algorithm that is proven to have any quadratic optimality. In this work, based on our GeneralACL, we propose **HyperACL**, which takes an input hypergraph with **e**dge-**d**ependent **v**ertex **w**eights (i.e., EDVW hypergraph, as defined in Definition 5). Inheriting from GeneralACL, HyperACL can, with at least $\frac{1}{2}$ probability, find a quadratically optimal local cluster in terms of hypergraph conductance (which is defined in this work, i.e., Definition 14).

## 1.1 Main Results

In this paper, we further extend the non-approximating Andersen-Chung-Lang ("ACL") algorithm (Andersen et al., 2006) to weighted, directed, self-looped graphs, and generalize its quadratic optimality. To enhance practicality, we introduce an early-stop mechanism that slightly sacrifices theoretical guarantees to maintain a strong locality. As a direct application, we study local clustering on hypergraphs. We show that, under two mild conditions, the seeded random walk can generate sweep cuts that produce a cluster with quadratically optimal conductance with at least $\frac{1}{2}$ probability, consistent with prior results (Andersen et al., 2008). The assumptions of the following two theorems are that (1) the input hypergraph should be connected (definition in (Li et al., 2024b)). Otherwise, there would be a trivial clustering solution that can minimize the conductance to 0; (2) the graph random walk is well-defined, and its stationary distribution exists. It has been proven that the stationary distribution exists for hypergraph random walks (Li et al., 2024b).

**Theorem 1.** *(For local clustering on graphs) Given any graph in the formatting $\mathcal{G} = (\mathcal{V}, \mathcal{W})$ with nonnegative edge weights $\mathcal{W}_{u,v} > 0$ for any $u \in \mathcal{V}, v \in \mathcal{V}$ and positive vertex weights $\phi_u(\cdot)$ for any $u \in \mathcal{V}$, for a vertex set $\mathcal{S} \subseteq \mathcal{V}$ that is uniformly randomly sampled from the subsets of $\mathcal{V}$, let $\mathcal{S}^*$ be the vertex set with optimal conductance among all the vertex sets containing $\mathcal{S}$. If these two conditions are satisfied: (1) $vol(\mathcal{S}^*) \leq \frac{1}{2}$ and (2) $\mathcal{S}^*$ is also the vertex set with optimal conductance among all the vertex sets containing $\mathcal{S}^* \setminus \mathcal{S}$; Then, there is an algorithm that can find a local cluster $\mathcal{S}'$ such that with probability at least $\frac{1}{2}$,*

$$\Phi(\mathcal{S}') < O(\sqrt{\Phi(\mathcal{S}^*)}) \tag{1}$$

**Theorem 2.** *(For local clustering on hypergraphs) Given any hypergraph in the EDVW formatting $\mathcal{H} = (\mathcal{V}, \mathcal{E}, \omega, \gamma)$ with positive edge weights $\omega(\cdot) > 0$ and non-negative edge-dependent vertex weights $\gamma_e(\cdot)$ for any hyperedge $e \in \mathcal{E}$, define the conductance of a vertex set as Definition 14. For a vertex set $\mathcal{S} \subseteq \mathcal{V}$ that is uniformly randomly sampled from the subsets of $\mathcal{V}$, let $\mathcal{S}^*$ be the vertex set with optimal conductance among all the vertex sets containing $\mathcal{S}$. If these two conditions are satisfied: (1) $vol(\mathcal{S}^*) \leq \frac{1}{2}$ and (2) $\mathcal{S}^*$ is also the vertex set with optimal conductance among all the vertex sets containing $\mathcal{S}^* \setminus \mathcal{S}$; Then, there is an algorithm that can find a local cluster $\mathcal{S}'$ such that with probability at least $\frac{1}{2}$,*

$$\Phi(\mathcal{S}') < O(\sqrt{\Phi(\mathcal{S}^*)}) \tag{2}$$

We propose GeneralACL and HyperACL. To the best of our knowledge, GeneralACL is the first local clustering algorithm for potentially weighted, directed, and self-looped graphs with quadratic optimality; HyperACL is the first hypergraph local clustering algorithm that satisfies the quadratically optimal guarantee, extending the non-approximating "ACL" (Andersen et al., 2006) algorithm to the EDVW hypergraphs.

**Technical Overview.** In this work, we address the challenges posed by generalizing clustering algorithms to weighted, directed, self-looped graphs and hypergraphs modeled with edge-dependent vertex weights (EDVW). While such structures allow more expressive modeling of real-world data, algorithmic developments for these settings remain limited, thereby constraining their broader applications. To bridge this gap, we extend the Andersen-Chung-Lang (ACL) algorithm to these generalized graphs and hypergraphs.

Random walks play a critical role in this work. We use the key insight from the previous work (Chitra & Raphael, 2019) to model the hypergraphs similar to a Markov process through the equivalence of random walks and key insight from the previous work (Andersen et al., 2008) to generalize the conductance of directed graphs using random walks. Our approach starts by re-analyzing random walks on both weighted directed graphs with self-loops and EDVW hypergraphs, unifying their local clustering formulations through Markov chains. This unification streamlines the subsequent proofs by eliminating the redundancy of repeating the same proof twice.

Our theoretical contributions culminate in Theorem 1, which generalizes the ACL algorithm to weighted directed graphs with self-loops, and Theorem 2, which adapts it to EDVW hypergraphs. We show that using our definitions on hypergraphs in EDVW formatting, we can also prove a quadratically optimal result, which is consistent with previous results (Andersen et al., 2006; 2008). The proof exploits the Lovász-Simonovits Curve. We first show that for a lazy PageRank vector $pr(\alpha, s)$ with stochastic vector $s$, the disagreement of the corresponding Lovász-Simonovits Curve and $f(x) = x$ is upper-bounded by a value associated with the smallest conductance of the sweep cuts. Then, we show that such disagreement also has a lower bound associated with the optimal conductance of any local cluster. Through these two inequalities, we connect the smallest conductance of the sweep cuts with the optimal conductance and obtain a quadratic relationship between them.

## 1.2 Paper Organization

This paper is organized as follows. Section 2 introduces the necessary notations, definitions, and assumptions for weighted directed self-looped graphs and hypergraphs with edge-dependent vertex weights. Section 3 leverages the Markov chain induced by random walks to present the conductance metric, provides an overview of PageRank and the sweep cut, and introduces our main results (Theorems 1 and 2). The complete proofs for these results are detailed in Section 4. Section 5 examines the complexity of our proposed algorithms. Finally, Section 6 presents comprehensive experiments to validate our theoretical findings. Our Appendix contains supplementary contents, such as trivial proofs and experimental details.

## 1.3 Other Related Works

**Local Clustering.** Local clustering is a fundamental task in data analytics (Berkhin, 2006), enabling efficient exploration and analysis of data subsets. Local graph clustering (Schaeffer, 2007) has been widely adopted across various data modalities by transforming raw data into graph structures (Shi & Malik, 2000). Many existing methods rely on random walks (Andersen et al., 2006; 2008; Avrachenkov et al., 2008) or diffusion processes (Chung, 2009; Chung & Simpson, 2018; Fountoulakis et al., 2020) to identify clusters based on local graph connectivity. In the era of big data and AI, local clustering techniques have evolved to address increasingly complex data settings (Zheng et al., 2024). These include noisy data (de Luca et al., 2024), attributed graphs (Yang & Fountoulakis, 2023), high-order interactions (Yin et al., 2017b;a; Fu et al., 2024a), and dynamic graphs (Fu et al., 2020; 2023). These advancements underscore the importance of local clustering as a versatile tool for analyzing diverse and complex datasets.

**Hypergraphs.** A variety of techniques for hypergraph analysis have been proposed (Gao et al., 2022; Antelmi et al., 2024; Lee et al., 2024), though only a limited number address EDVW hypergraphs, whose spectral properties have been recently studied in (Li et al., 2024b). On the local clustering front, LQHD

Table 1: Table of Notation

| Symbol | Definition and Description |
|---|---|
| $\mathcal{G} = (\mathcal{V}, W, \varphi)$ | graph being investigated, with vertex set $\mathcal{V}$, adjacency matrix $W$, vertex weight mapping $\varphi$ |
| $\mathcal{H} = (\mathcal{V}, \mathcal{E}, \omega, \gamma)$ | hypergraph being investigated, with vertex set $\mathcal{V}$, hyperedge set $\mathcal{E}$, edge weight mapping $\omega$ and edge-dependent vertex weight mapping $\gamma$ |
| $n = |\mathcal{V}|$ | number of vertices |
| $m$ | number of hyperedge-vertex connections in Hypergraph $\mathcal{H}$, $m = \sum_{e \in \mathcal{E}} |e|$ |
| $d(v)$ | degree of vertex $v$, $d(v) = \sum_{e \in E(v)} w(e)$ |
| $\delta(e)$ | degree of hyperedge $e$, $\delta(e) = \sum_{v \in e} \gamma_e(v)$ |
| $R$ | $|\mathcal{E}| \times |\mathcal{V}|$ vertex-weight matrix |
| $W$ | $|\mathcal{V}| \times |\mathcal{E}|$ hyperedge-weight matrix |
| $D_{\mathcal{V}}$ | $|\mathcal{V}| \times |\mathcal{V}|$ vertex-degree matrix |
| $D_{\mathcal{E}}$ | $|\mathcal{E}| \times |\mathcal{E}|$ hyperedge-degree matrix |
| $P$ | $|\mathcal{V}| \times |\mathcal{V}|$ transition matrix of random walk on $\mathcal{H}$ |
| $\phi$ | $1 \times |\mathcal{V}|$ stationary distribution of random walk |
| $\Pi$ | $|\mathcal{V}| \times |\mathcal{V}|$ diagonal stationary distribution matrix |
| $p$ | $1 \times |\mathcal{V}|$ probability distribution on $\mathcal{V}$ |
| $\alpha$ | restart probability of random walks |
| $\partial$ | boundary notation |
| $vol$ | volume notation |
| $pr(\alpha, s)$ | lazy personalized PageRank vector with restart probability $\alpha$ and stochastic vector $s$ |

(local hypergraph quadratic diffusions)(Liu et al., 2021) introduced a strongly local diffusion algorithm for non-EDVW hypergraphs, focusing on semi-supervised community recovery. However, LQHD lacks any proof of optimality, limiting its theoretical robustness. Other works (Zhong et al., 2023; Ibrahim & Gleich, 2020; Prokopchik et al., 2022; Ibrahim & Gleich, 2019; Fountoulakis et al., 2021; Wang et al., 2023a) explored alternative diffusion approaches for non-EDVW hypergraphs. Additionally, several studies have investigated specific applications of hypergraph clustering, such as community detection and modularization (Kamhoua et al., 2021; Chang et al., 2023). DiffEq (Takai et al., 2020) formulates hypergraph clustering using a differential equation framework and employs a distinct sweep cut approach compared to ours. In our experiments, we demonstrate that HyperACL outperforms DiffEq.

**PageRank.** PageRank (Page et al., 1999), playing a crucial role in ranking systems, has been widely applied in graph analysis and mining, especially for large networks. Some recent advances in this algorithm include theoretical optimality (Wang et al., 2024), more accelerated computation (Li et al., 2023; Zhang et al., 2023; Yang et al., 2024) and applications on diverse tasks (Ban et al., 2024b; Stoica et al., 2024). Despite these advancements, most approaches focus on simple graph structures, overlooking more complex settings, as in this work.

## 2 Preliminaries

We use calligraphic letters (e.g., $\mathcal{A}$) for sets, capital letters for matrices (e.g., $A$), and unparenthesized superscripts to denote the power (e.g., $A^k$). For matrix indices, we use $A_{i,j}$ or $A(i,j)$ interchangeably to denote the entry in the $i^{th}$ row and the $j^{th}$ column. For row vector or column vector $v$, we use $v(i)$ to index its $i^{th}$ entry. Also, we denote graph as $\mathcal{G}$ and hypergraph as $\mathcal{H}$.

## 2.1 Graphs, Possibly Edge-weighted, Node-weighted, Directed and Self-looped

**Definition 3.** *(Graph). A graph $\mathcal{G} = (\mathcal{V}, W, \varphi)$ is defined as a set of vertices $\mathcal{V}$, a (generalized) adjacency matrix $W \in \mathbb{R}_{\geq 0}^{|\mathcal{V}| \times |\mathcal{V}|}$, vertex weights $\varphi(v) : \mathcal{V} \to \mathbb{R}_+$ on every vertex $v \in \mathcal{V}$. Without loss of generality, we index the vertices by $1, 2, ..., |\mathcal{V}|$, and let $\mathcal{V} = \{1, 2, ..., |\mathcal{V}|\}$.*

For typical graphs, all edge weights and node weights are equal to one. In this work, we allow the graphs to have diverse edge weights and node weights, enabling a more general representation of real-world systems. Specifically, we allow the entries of the adjacency matrix $W$ to represent non-negative weights assigned to edges, where $W_{i,j} = 0$ indicates the absence of an edge between vertices $i$ and $j$. These edge weights can model various properties such as capacities, distances, or affinities between connected vertices.

Similarly, the vertex weights $\varphi(v)$ are generalized to capture the importance, influence, or other attributes of individual vertices in the graph. For instance, vertex weights may represent node-specific features such as population in a geographic network, processing power in a computing network, or centrality in a social network.

A graph is **directed** if and only if its adjacency matrix $W$ is not symmetric, meaning there exists $i, j \in \mathcal{V}$ such that $W_{i,j} \neq W_{j,i}$. In this case, the direction of the edge is significant, representing an edge from vertex $i$ to vertex $j$ but not necessarily vice versa. Directed graphs are commonly used in applications such as modeling flows, ranking systems, or dependency relationships.

A graph is **self-looped** if and only if there exists at least one vertex $v \in \mathcal{V}$ such that $W_{v,v} > 0$. In this scenario, the diagonal entries of the adjacency matrix are non-zero for such vertices, indicating the presence of self-loops. Self-loops are relevant in scenarios where an entity is directly connected to itself, such as feedback systems or certain types of network structures.

In the discrete graph setting, diverse edge weights, diverse node weights, directions, and self-loops are not allowed. In other words, $W$ only contains 0 or 1; $\varphi(\cdot)$ is constantly 1; $W$ must be symmetric and the diagonal elements of $W$ are 0. Which limits the expressiveness and applicability of such graphs in modeling complex systems. Our relaxations and generalizations enable more flexible and nuanced modeling of complex systems, broadening the applicability across diverse domains, including transportation, communication, biology, and social sciences.

Without loss of generality, we assume the graph is strongly connected. General connectivity is essential; otherwise, there is a trivial solution for clustering. If the graph is not strongly connected in practice, the methods can be applied to each strongly connected component independently.

**Definition 4.** *(Chitra & Raphael, 2019) (Graph random walk). A random walk on a graph is a Markov Chain on $\mathcal{V}$ with transition probabilities matrix $P$. The stationary distribution of the random walk with transition matrix $P$ is a $1 \times |\mathcal{V}|$ row vector $\phi$ such that*

$$\phi P = \phi; \ \phi(u) > 0 \ \forall u \in \mathcal{V}; \ \sum_{u \in \mathcal{V}} \phi(u) = 1 \tag{3}$$

For discrete graph setting, the transition matrix $P = D^{-1}W$ where $D$ is the diagonal degree matrix $D_{i,i} = \sum_{k=1}^{|\mathcal{V}|} W_{i,k}$. However, for a more generalized graph setting, the construction of transition matrix $P$ from the edge and node weights depends on the specific data and task domain, which is beyond the scope of this work. While the specific construction of $P$ for generalized graphs may vary, this work focuses on leveraging the properties of $P$ and $\pi$ for algorithmic design and theoretical analysis. Without loss of generality, we assume there is a well-defined irreducible and aperiodic transition matrix $P$ that satisfies the properties of a row-stochastic matrix, i.e., each row of $P$ sums to 1, and all entries $P_{i,j} \geq 0$.

Furthermore, we assume the existence of a unique stationary distribution $\pi$ associated with $P$. These properties are essential for ensuring convergence to a steady state, making $\pi$ well-defined and meaningful for clustering and other downstream tasks.

**Assumption 1.** *We assume the existence of a well-defined, irreducible, and aperiodic transition matrix $P$ that is row-stochastic, with each row summing to 1 and all entries $P_{i,j} \geq 0$. Additionally, we assume the existence of a unique stationary distribution $\pi$ satisfying $\pi P = \pi$.*

## 2.2 Hypergraphs, with Possibly Edge-dependent Vertex Weights

A hypergraph consists of vertices and hyperedges. A hyperedge $e$ is a connection between two or more vertices. We use the notation $v \in e$ if the hyperedge $e$ connects vertex $v$. This is also called "$e$ is incident to $v$". Edge-dependent vertex weights (EDVW) modeling (Chitra & Raphael, 2019) is one of the most generalized modeling methods of hypergraphs. We first provide the formal definition of an EDVW hypergraph. Definition 5 and 6 provide necessary notations to define the hypergraph random walk in Definition 7. The transition matrix $P$ of EDVW hypergraphs is consistent with that of graphs.

**Definition 5.** *(Chitra & Raphael, 2019) (EDVW hypergraph). A hypergraph $\mathcal{H} = (\mathcal{V}, \mathcal{E}, \omega, \gamma)$ with edge-dependent vertex weight is defined as a set of vertices $\mathcal{V}$, a set $\mathcal{E} \subseteq 2^{\mathcal{V}}$ of hyperedges, a weight mapping $\omega(e) : \mathcal{E} \to \mathbb{R}_+$ on every hyperedge $e \in \mathcal{E}$, and weight mappings $\gamma_e(v) : \mathcal{V} \to \mathbb{R}_{\geq 0}$ corresponding to $e$ on every vertex $v$. For $e_1 \neq e_2$, $\gamma_{e_1}(v)$ and $\gamma_{e_2}(v)$ may be different. Without loss of generality, we index the vertices by $1, 2, ..., |\mathcal{V}|$, and let $\mathcal{V} = \{1, 2, ..., |\mathcal{V}|\}$.*

For an EDVW hypergraph $\mathcal{H} = (\mathcal{V}, \mathcal{E}, \omega, \gamma)$, $\omega(e) > 0$ for any $e \in \mathcal{E}$. $\gamma_e(v) \geq 0$ for any $e \in \mathcal{E}$ and $v \in \mathcal{V}$. Moreover, $\gamma_e(v) > 0 \iff v \in e$. For instance, in a citation hypergraph, each publication is captured by a hyperedge. While each publication may have different citations (i.e., edge-weight $w(e)$), each author may have individual weight of contributions (i.e., publication-dependent $\gamma_e(v)$).

**Definition 6.** *(Chitra & Raphael, 2019) (Vertex-weight matrix, hyperedge-weight matrix, vertex-degree matrix, and hyperedge-degree matrix of an EDVW hypergraph). $E(v) = \{e \in \mathcal{E} \ s.t. \ v \in e\}$ is the set of hyperedges incident to vertex $v$. $d(v) = \sum_{e \in E(v)} w(e)$ denotes the degree of vertex $v$. $\delta(e) = \sum_{v \in e} \gamma_e(v)$ denotes the degree of hyperedge $e$. The vertex-weight matrix $R$ is an $|\mathcal{E}| \times |\mathcal{V}|$ matrix with entries $R(e, v) = \gamma_e(v)$. The hyperedge-weight matrix $W$ is a $|\mathcal{V}| \times |\mathcal{E}|$ matrix with entries $W(v, e) = \omega(e)$ if $v \in e$, and $W(v, e) = 0$ otherwise. The vertex-degree matrix $D_{\mathcal{V}}$ is a $|\mathcal{V}| \times |\mathcal{V}|$ diagonal matrix with entries $D_{\mathcal{V}} = d(v)$. The hyperedge-degree matrix $D_{\mathcal{E}}$ is a $|\mathcal{E}| \times |\mathcal{E}|$ diagonal matrix with entries $D_{\mathcal{E}}(e, e) = \delta(e)$.*

**Assumption 2.** *Without loss of generality, we assume the hypergraph is connected. A rigorous definition of hypergraph connectivity is provided in (Zhou et al., 2006) and (Li et al., 2024b).*

Table 1 contains important notation and hyperparameters for quick reference. Then, following previous work (Chitra & Raphael, 2019; Li et al., 2024b), we introduce hypergraph random walk. To maintain consistency, we use the same notation as graph random walks, as both are fundamentally Markov processes.

**Definition 7.** *(Chitra & Raphael, 2019) (Hypergraph random walk). A random walk on a hypergraph with edge-dependent vertex weights $\mathcal{H} = (\mathcal{V}, \mathcal{E}, \omega, \gamma)$ is a Markov Chain on $\mathcal{V}$ with transition probabilities*

$$P_{u,v} = \sum_{e \in E(u)} \frac{\omega(e)}{d(u)} \frac{\gamma_e(v)}{\delta(e)} \tag{4}$$

*$P$ can be written in matrix form as $P = D_{\mathcal{V}}^{-1} W D_{\mathcal{E}}^{-1} R$ (Chitra & Raphael, 2019) and it has row sum of 1 (Li et al., 2024b).*

**Definition 8.** *(Stationary distribution of hypergraph random walk). The stationary distribution of the random walk with transition matrix $P$ is a $1 \times |\mathcal{V}|$ row vector $\phi$ such that*

$$\phi P = \phi; \ \phi(u) > 0 \ \forall u \in \mathcal{V}; \ \sum_{u \in \mathcal{V}} \phi(u) = 1 \tag{5}$$

*The existence of stationary distribution $\phi$ has been proved in (Chitra & Raphael, 2019) and (Li et al., 2024b). From $\phi$, we further define the stationary distribution matrix to be a $|\mathcal{V}| \times |\mathcal{V}|$ diagonal matrix with entries $\Pi_{i,i} = \phi(i)$.*

From Definition 9 to Definition 12, in the context of EDVW hypergraph, following the definitions on graphs, we re-define the volume of boundaries and vertex sets, which are crucial for local clustering. We show that our definitions have properties that are consistent with those on graphs. Given a vertex set $\mathcal{S} \subseteq \mathcal{V}$, we use $\bar{\mathcal{S}}$ to denote its *complementary set*, where $\mathcal{S} \cup \bar{\mathcal{S}} = \mathcal{V}$ and $\mathcal{S} \cap \bar{\mathcal{S}} = \emptyset$. We have the following definition regarding the probability of a set.

**Definition 9.** *(Probability of a set). For a distribution $p$ on the vertices such that $\forall v \in \mathcal{V}, p(v) \geq 0$ and $\sum_{v \in \mathcal{V}} p(v) = 1$, we denote*

$$p(S) = \sum_{x \in S} p(x), \forall S \subseteq V \tag{6}$$

Equivalently, we can regard $p$ as a $1 \times |\mathcal{V}|$ vector with $p_i = p(i)$. From this definition, $\phi(\mathcal{S}) + \phi(\bar{\mathcal{S}}) = 1$. By definition of random walk and its stationary distribution,

$$\begin{aligned}\phi(S) &= (\phi P)(S) \\ &= \phi(S) - \sum_{u \in \mathcal{S}, v \in \bar{\mathcal{S}}} \phi(u) P_{u,v} + \sum_{u \in \bar{\mathcal{S}}, v \in \mathcal{S}} \phi(u) P_{u,v}\end{aligned} \tag{7}$$

Therefore, we have the following Theorem 10 that, in the stationary state, for any set $\mathcal{S}$, the probability of walking into $\mathcal{S}$ or out of $\mathcal{S}$ are the same (rigorous proof in Lemma 39).

**Theorem 10.** *Let $\mathcal{H} = (\mathcal{V}, \mathcal{E}, \omega, \gamma)$ be a hypergraph with edge-dependent vertex weights. Let $P$ be the transition matrix and $\phi$ be the corresponding stationary distribution. Then, for any vertex set $\mathcal{S} \subseteq \mathcal{V}$,*

$$\sum_{u \in \mathcal{S}, v \in \bar{\mathcal{S}}} \phi(u) P_{u,v} = \sum_{u \in \bar{\mathcal{S}}, v \in \mathcal{S}} \phi(u) P_{u,v} \tag{8}$$

For unweighted and undirected graphs, the volume of the *boundary/cut* of a partition is defined as $|\partial \mathcal{S}| = |\{\{x, y\} \in E | x \in \mathcal{S}, y \in \bar{\mathcal{S}}\}|$. The intuition behind this definition is the symmetric property $|\partial \mathcal{S}| = |\partial \bar{\mathcal{S}}|$. Theorem 10 also describes such a property, and we find it intuitively suitable to be extended to the following definition.

**Definition 11.** *(Volume of hypergraph boundary). We define the volume of the hypergraph boundary, i.e., cut between $\mathcal{S}$ and $\bar{\mathcal{S}}$, by*

$$|\partial S| = \sum_{u \in \mathcal{S}, v \in \bar{\mathcal{S}}} \phi(u) P_{u,v} = \sum_{u \in \bar{\mathcal{S}}, v \in \mathcal{S}} \phi(u) P_{u,v} \tag{9}$$

*Furthermore, $0 \leq |\partial \mathcal{S}| \leq \sum_{u \in \mathcal{S}} \phi(u) \leq 1$. $|\partial \mathcal{S}| = |\partial \bar{\mathcal{S}}|$.*

For unweighted, undirected graphs, the *volume of a vertex set $\mathcal{S}$* is defined as the degree sum of the vertices in $\mathcal{S}$. With the observation that $\phi(u) = \sum_{u \in \mathcal{S}, v \in \mathcal{V}} \phi(u) P_{u,v}$, $\phi(u)$ itself is already a sum of the transition probabilities and can be an analogy to vertex degree. We extend this observation to the following definition.

**Definition 12.** *(Volume of hypergraph vertex set). We define the volume of a vertex set $\mathcal{S} \subseteq \mathcal{V}$ in hypergraph $\mathcal{H}$ by*

$$vol(S) = \sum_{u \in \mathcal{S}} \phi(u) \in [0, 1] \tag{10}$$

Furthermore, we have $vol(\emptyset) = 0$, $vol(\mathcal{V}) = 1$, and $vol(\mathcal{S}) + vol(\bar{\mathcal{S}}) = 1$. Definition 11 and Definition 12 will serve as the basis of our unified formulation. By these two definitions, we also have $|\partial \mathcal{S}| \leq vol(\mathcal{S})$, which is consistent with those on unweighted and undirected graphs.

## 3 Local Clustering

In this section, we first extend conductance, a widely accepted measurement of local cluster qualities, for general graphs and EDVW hypergraphs. We then introduce our GeneralACL (Algorithm 1) and HyperACL (Algorithm 2), which heuristically sweeps over a set of highly promising candidates to be included in the local cluster.

### 3.1 Conductance Measures as Markov Chains

The conductance measures the proportion of "broken" edges relative to the total importance of the two clusters. It quantifies the quality of a cluster by balancing the separation between clusters and their internal connectivity. The measurement of local cluster quality requires the generalization of conductance metric as well. Following the previous work (Andersen et al., 2008; 2006), we present a generalized definition of conductance for graphs that may be edge-weighted, node-weighted, directed, and self-looped. This extends the original definition in (Andersen et al., 2008) to accommodate more complex graph structures.

**Definition 13.** *(General graph conductance). The conductance of a cluster $\mathcal{S}$ on $\mathcal{G}$ with transition matrix $P$ and stationary distribution $\phi$ is,*

$$\Phi_{\mathcal{G}}(\mathcal{S}) = \frac{\sum_{u \in \mathcal{S}, v \in \bar{\mathcal{S}}} \phi(u) P_{u,v}}{\min(\sum_{v \in \mathcal{S}} \phi(u), 1 - \sum_{u \in \mathcal{S}} \phi(u))} \tag{11}$$

In this work, we follow the conventional formulation $\Phi(\mathcal{S}) = \frac{|\partial \mathcal{S}|}{\min(vol(\mathcal{S}), vol(\bar{\mathcal{S}}))}$ for conductance but applied to hypergraphs in EDVW formatting for the first time.

**Definition 14.** *(Hypergraph conductance). The conductance of a cluster $\mathcal{S}$ on $\mathcal{H}$ with transition matrix $P$ and stationary distribution $\phi$ is,*

$$
\begin{aligned}
\Phi_{\mathcal{H}}(\mathcal{S}) &= \frac{|\partial \mathcal{S}|}{\min(vol(\mathcal{S}), vol(\bar{\mathcal{S}}))} \\
&= \frac{|\partial \mathcal{S}|}{\min(vol(\mathcal{S}), 1 - vol(\mathcal{S}))} \\
&= \frac{\sum_{u \in \mathcal{S}, v \in \bar{\mathcal{S}}} \phi(u) P_{u,v}}{\min(\sum_{v \in \mathcal{S}} \phi(u), 1 - \sum_{u \in \mathcal{S}} \phi(u))}
\end{aligned}
\tag{12}
$$

**Theorem 15.** *For any vertex set $\mathcal{S} \subseteq \mathcal{V}$, our hypergraph conductance $\Phi_{\mathcal{H}}(\mathcal{S}) \in [0, 1]$, which is consistent with graph conductance*

*Proof.* Recall from the definition 14, 11 and 12,

$$\Phi_{\mathcal{H}}(\mathcal{S}) = \frac{|\partial \mathcal{S}|}{\min(vol(\mathcal{S}), vol(\bar{\mathcal{S}}))} = \frac{\sum_{u \in \mathcal{S}, v \in \bar{\mathcal{S}}} \phi(u) P_{u,v}}{\min(\sum_{v \in \mathcal{S}} \phi(u), \sum_{u \in \bar{\mathcal{S}}} \phi(u))} \tag{13}$$

From the definition of $P_{u,v}$ and $\phi(u)$, $\Phi(\mathcal{S})$ is non-negative.

$$
\begin{aligned}
\sum_{u \in \mathcal{S}} \phi(u) &= \sum_{u \in \mathcal{S}, v \in \mathcal{V}} \phi(u) P_{u,v} \geq \sum_{u \in \mathcal{S}, v \in \bar{\mathcal{S}}} \phi(u) P_{u,v} \\
\sum_{u \in \bar{\mathcal{S}}} \phi(u) &= \sum_{u \in \bar{\mathcal{S}}, v \in \mathcal{V}} \phi(u) P_{u,v} \geq \sum_{u \in \bar{\mathcal{S}}, v \in \mathcal{S}} \phi(u) P_{u,v} \overset{Equation\ 10}{=} \sum_{u \in \mathcal{S}, v \in \bar{\mathcal{S}}} \phi(u) P_{u,v}
\end{aligned}
\tag{14}
$$

Thus, $\min(\sum_{v \in \mathcal{S}} \phi(u), \sum_{u \in \bar{\mathcal{S}}} \phi(u)) \geq \sum_{u \in \mathcal{S}, v \in \bar{\mathcal{S}}} \phi(u) P_{u,v}$ and $\Phi_{\mathcal{H}}(\mathcal{S}) \in [0, 1]$. $\qquad\square$

Notably, the conductance of general graphs and hypergraphs shares the same formulation on their equivalent Markov chains. The later proofs leveraging conductance require no additional properties beyond those inherent to the corresponding Markov chains. Therefore, we unify the notation and use $\Phi$ to represent both $\Phi_{\mathcal{G}}$ (for graphs) and $\Phi_{\mathcal{H}}$ (for hypergraphs).

### 3.2 GeneralACL and HyperACL Algorithms based on Personalized PageRank

Given a starting vertex set, finding optimal local clusters in terms of minimal conductance is an NP-complete problem (Sima & Schaeffer, 2006). GeneralACL and HyperACL work by ranking the vertex candidacy heuristically using seeded/personalized random walks.

**Definition 16.** *(Lazy PPR vector) The lazy personalized PageRank vector $pr(\alpha, s)$ on $\mathcal{H}$ is defined as follows.*

$$pr(\alpha, s) = \alpha s + (1 - \alpha)pr(\alpha, s)M \tag{15}$$

*where $M = \frac{1}{2}(I + P)$, $s$ is the stochastic vector of random walks, $\alpha$ is the restart probability of random walk. The existence of $pr(\alpha, s)$ can be proved by verifying $pr(\alpha, s) = \alpha s \sum_{k=0}^{\infty}((1 - \alpha)M)^k$ (Li et al., 2023). In fact, a lazy random walk is equivalent to a standard random walk by a switch of $\alpha$, as the lemma below states.*

**Lemma 17.** *Denote $rpr(\alpha', s)$ to be the solution of*

$$rpr(\alpha', s) = \alpha' s + (1 - \alpha')rpr(\alpha', s)P \tag{16}$$

*then $pr(\alpha, s) = rpr(\frac{2\alpha}{1+\alpha}, s)$. (Proof in Appendix A.1)*

---

**Algorithm 1** GeneralACL

**Require:** Graph $\mathcal{G} = (\mathcal{V}, W, \varphi)$; a set of starting vertices $\mathcal{S}$ for local clustering
**Ensure:** a cluster of vertices
 1: Compute the transition matrix $P$ as assumption 1.
 2: Compute the stationary distribution by power iteration.
 3: Compute the PageRank vector as in Theorem 21.
 4: Compute the sweep sets by Definition 19.
 5: Compute which sweep set has the smallest conductance.
 6: Return the sweep set with the smallest conductance.

---

**Algorithm 2** HyperACL

**Require:** EDVW hypergraph $\mathcal{H} = (\mathcal{V}, \mathcal{E}, \omega, \gamma)$; a set of starting vertices $\mathcal{S}$ for local clustering
**Ensure:** a cluster of vertices
 1: Compute $R, W, D_\mathcal{V}, D_\mathcal{E}$ according to Definition 6.
 2: Compute the transition matrix $P$ by Definition 7.
 3: Compute the stationary distribution by power iteration.
 4: Compute the PageRank vector as in Theorem 21.
 5: Compute the sweep sets by Definition 19.
 6: Compute which sweep set has the smallest conductance.
 7: Return the sweep set with the smallest conductance.

---

**Definition 18.** *The indicator function $\chi_v$ is defined as*

$$\chi_v(x) = \begin{cases} 1 & x = v \\ 0 & otherwise \end{cases} \tag{17}$$

*and we denote $pr(\alpha, \chi_v)$ to be the lazy single-source PageRank vector with respect to $v$.*

Sweep sets serve as an effective manner for obtaining the local clustering as follows.

**Definition 19.** *(Andersen et al., 2006) (Sweep sets/cuts). For a distribution $p$ on the vertices such that $\forall v \in \mathcal{V}, p(v) \geq 0; \sum_{v \in \mathcal{V}} p(v) = 1$, let $N_p = |Supp(p)|$ be the support size (a.k.a., number of vertices $v$ with $p(v) \neq 0$), and let $v_1, v_2, ..., v_{N_p}$ be an ordering of vertices such that $\frac{p(v_i)}{\phi(v_i)} \geq \frac{p(v_{i+1})}{\phi(v_{i+1})}$. From this ordering we can obtain $N_p$ vertex sets $\mathcal{S}_1^p, \mathcal{S}_2^p, ..., \mathcal{S}_{N_p}^p$ such that $\mathcal{S}_j^p = \{v_1, v_2, ..., v_j\}, \forall j \in [1, N_p]$. We call the ordered sets $\mathcal{S}_1^p, \mathcal{S}_2^p, ..., \mathcal{S}_{N_p}^p$ sweep sets or sweep cuts. Each $\mathcal{S}_j^p$ is called a sweep set/cut.*

Directly from the above definition, we have $p(\mathcal{S}_j^p) \geq 0$, $p(\mathcal{S}_j^p) < p(\mathcal{S}_{j+1}^p)$, $p(\mathcal{S}_{N_p}^p) = 1$. According to the monotonicity of $\frac{p(v_i)}{\phi(v_i)}$, we have $\frac{p(\mathcal{S}_j^p)}{vol(\mathcal{S}_j^p)} \geq \frac{p(\mathcal{S}_{j+1}^p)}{vol(\mathcal{S}_{j+1}^p)}$ $\forall j \in [1, N_p-1]$; and $\frac{p(\mathcal{S}_j^p)}{vol(\mathcal{S}_j^p)} \geq \frac{p(\mathcal{S})}{vol(\mathcal{S})}$ $\forall \mathcal{S} \subseteq |V|$, $|\mathcal{S}| = |\mathcal{S}_j^p|$.

**Definition 20.** *(Andersen et al., 2006) (Optimal conductance of a distribution) We denote the optimal conductance of a distribution to be the smallest conductance of any of its sweep sets.*

$$\Phi(p) = \min_{j \in [1, N_p]} \Phi(\mathcal{S}_j^p) \tag{18}$$

*which can be found by sorting $\frac{p(v_i)}{\phi(v_i)}$ and computing the conductance of each sweep set.*

Given seed vertices, we calculate the random walk probability starting from these seed vertices, a.k.a. PageRank vector, and the corresponding sweep. The vertices that are ranked in the front of the sweep have high seeded-random-walk distributions, weighted by the stationary random walk distributions. Heuristically, this means the vertices that are ranked in the front are relatively more closely connected to the seed vertices. We prove the following theorem, which shows that GeneralACL and HyperACL, leveraging the Markov chains induced by random walks on either general graph or hypergraph, in many situations, find a quadratically optimal local cluster in terms of conductance.

**Theorem 21.** *For a vertex set $\mathcal{S} \subseteq \mathcal{V}$ that is uniformly randomly sampled from the subsets of $\mathcal{V}$, let $\mathcal{S}^*$ be the vertex set with optimal conductance among all the vertex sets containing $\mathcal{S}$. Let the PageRank Starting Distribution be*

$$\psi_{\mathcal{S}}(v) = \begin{cases} \frac{\phi(v)}{vol(\mathcal{S})} & , \ v \in \mathcal{S} \\ 0 & , \ otherwise \end{cases} \tag{19}$$

*and let $\alpha = \Phi(\mathcal{S}^*) \in [0,1]$. If these two conditions are satisfied, i.e., (1) $vol(\mathcal{S}^*) \leq \frac{1}{2}$ and (2) $\mathcal{S}^*$ is also the vertex set with optimal conductance among all the vertex sets containing $\mathcal{S}^* \setminus \mathcal{S}$, then with probability[2] at least $\frac{1}{2}$, we have*

$$\Phi(pr(\alpha, \psi_{\mathcal{S}})) < O(\sqrt{\Phi(\mathcal{S}^*)}) \tag{20}$$

*Specifically, when $vol(\mathcal{S}^*) \leq \frac{1}{3}$, then*

$$\Phi(pr(\alpha, \psi_{\mathcal{S}})) < \sqrt{235\Phi(\mathcal{S}^*)} \tag{21}$$

*(Proof in Section 4.5)*

It is worth noting that the most time-consuming step in GeneralACL and HyperACL is checking the conductance of every sweep set. We adopt an early-stop mechanism here to accelerate the computation: when sweeping over the candidate local clusters, if the conductance does not break the minimum within a manually set hyperparameter, then we assume we have obtained a local cluster that is good enough in terms of conductance. The hyperparameter can be adjusted to balance efficiency and accuracy.

## 4 Extending Andersen-Chung-Lang Algorithm

In this section, we aim to prove Theorem 21. We give the first proof that extends the famous Andersen-Chung-Lan Algorithm to edge-weighted and node-weighted directed graphs with self-loops and to hypergraphs with edge-dependent vertex weights. Still, we leverage the Lovász-Simonovits Curve. Unlike the original proof on unweighted undirected graphs, we cannot simply use the integer node degrees to define the volume of a vertex set. In our proof, the support of Lovász-Simonovits Curve is $[0, 1]$ instead of $[0, 2|\mathcal{E}|]$.

### 4.1 Lovász-Simonovits Curve

Lovász-Simonovits Curve (LSC or L-S Curve) serves as an important tool for our proofs, which defines a piecewise function as follows.

---

[2]the randomness comes from the sampling of $\mathcal{S}$

**Definition 22.** *Given a hypergraph $\mathcal{H} = (\mathcal{V}, \mathcal{E}, \omega, \gamma)$, a distribution $p$ and their corresponding sweep sets $\mathcal{S}_1^p, \mathcal{S}_2^p, ..., \mathcal{S}_{N_p}^p$, Lovász-Simonovits Curve defines a piecewise function $I_p : [0,1] \to [0,1]$ such that $\forall k \in [0,1]$, using $p[k]$ for short,*

$$p[k] = I_p(k) = \begin{cases} 0 & k = 0 \\ p(\mathcal{S}_j^p) & if \ \exists j \ s.t. \ k = vol(\mathcal{S}_j^p) \\ p(\mathcal{S}_j^p) + (k - vol(\mathcal{S}_j^p))\frac{p(v_{j+1})}{\phi(v_{j+1})} & if \ \exists j \ s.t. \ vol(\mathcal{S}_j^p) < k < vol(\mathcal{S}_{j+1}^p) \\ 1 & vol(\mathcal{S}_{N_p}^p) < k \le 1 \end{cases} \tag{22}$$

$p[k]$ is continuous because

when $k = vol(\mathcal{S}_j^p), p(\mathcal{S}_j^p) + (k - vol(\mathcal{S}_j^p))\frac{p(v_{j+1})}{\phi(v_{j+1})} = p(\mathcal{S}_j^p)$

when $k = vol(\mathcal{S}_{j+1}^p), p(\mathcal{S}_j^p) + (k - vol(\mathcal{S}_j^p))\frac{p(v_{j+1})}{\phi(v_{j+1})} = vol(\mathcal{S}_{j+1}^p) = p(\mathcal{S}_j^p) + \phi(v_{j+1})\frac{p(v_{j+1})}{\phi(v_{j+1})} = p(\mathcal{S}_{j+1}^p)$ (23)

when $k = vol(\mathcal{S}_{N_p}^p), p(\mathcal{S}_{N_p}^p) = 1$

Furthermore, $p[k]$ is concave because $\frac{p(\mathcal{S}_j^p)}{vol(\mathcal{S}_j^p)} \ge \frac{p(\mathcal{S}_{j+1}^p)}{vol(\mathcal{S}_{j+1}^p)} \ \forall j \in [1, N_p - 1]$.

The following definitions and lemmas about the Lovász-Simonovits Curve are important for later proofs.

**Definition 23.** *For any node pair $u, v \in \mathcal{V}$ (it is possible that $u = v$), we denote $p(u, v) = \frac{p(u)}{\phi(u)} P_{u,v}$.*

**Definition 24.** *For any edge/node-pair set $\mathcal{A}$, we denote $p(\mathcal{A}) = \sum_{(u,v) \in \mathcal{A}} \phi(u) p(u, v) = \sum_{(u,v) \in \mathcal{A}} p(u) P_{u,v}$.*

**Definition 25.** *For any edge/node-pair set $\mathcal{A}$, we denote $|\mathcal{A}| = \sum_{(u,v) \in \mathcal{A}} \phi(u) P_{u,v}$.*

Note that in the above definitions, the edge set $\mathcal{A} \in \mathcal{V} \times \mathcal{V}$ is not a hyperedge set, but only contains node pairs. Also, for an edge set $\mathcal{A}$, $|\mathcal{A}|$ in this paper does not refer to the cardinal number of $\mathcal{A}$.

**Lemma 26.** $\forall \mathcal{S} \subseteq \mathcal{V}, p(\mathcal{S}) \le p[vol(\mathcal{S})]$.

*Proof.* Recall from the definition 22, we can have this format of representation of L-S Curve:

$$p[vol(\mathcal{S})] = \begin{cases} 0 & vol(\mathcal{S}) = 0 \\ p(\mathcal{S}_j^p) & if \ \exists j \ s.t. \ vol(\mathcal{S}) = vol(\mathcal{S}_j^p) \\ p(\mathcal{S}_j^p) + (vol(\mathcal{S}) - vol(\mathcal{S}_j^p))\frac{p(v_{j+1})}{\phi(v_{j+1})} & if \ \exists j \ s.t. \ vol(\mathcal{S}_j^p) < vol(\mathcal{S}) < vol(\mathcal{S}_{j+1}^p) \\ 1 & vol(\mathcal{S}_{N_p}^p) < vol(\mathcal{S}) \le 1 \end{cases} \tag{24}$$

when $vol(\mathcal{S}) = 0$, the inequality holds as $p(\mathcal{S}) = p[vol(\mathcal{S})] = 0$;

when $vol(\mathcal{S}) = vol(\mathcal{S}_j^p)$, then $p(\mathcal{S}) = p(\mathcal{S}_j^p)$, so the inequality holds as $p(\mathcal{S}) = p[vol(\mathcal{S})] = p(\mathcal{S}_j^p)$ for some $j$;

when $vol(\mathcal{S}_{N_p}^p) < vol(\mathcal{S}) \le 1$, we can verify the inequality from the definition of $p$, as $p(\mathcal{S}) \le 1$, so the inequality still holds.

For $vol(\mathcal{S}_j^p) < vol(\mathcal{S}) < vol(\mathcal{S}_{j+1}^p)$, we can notice that $p[vol(\mathcal{S})]$ is a linear function of $vol(\mathcal{S})$ with slope $\frac{p(v_{j+1})}{\phi(v_{j+1})}$. Recall from the definition, we have proved that $p[k]$ is a continuous function. And 2 endpoints for the segment are $(vol(\mathcal{S}_j^p), p(\mathcal{S}_j^p))$ and $(vol(\mathcal{S}_{j+1}^p), p(\mathcal{S}_{j+1}^p))$, since $p(\mathcal{S})$ has no corresponding sweep set when $vol(\mathcal{S}_j^p) < vol(\mathcal{S}) < vol(\mathcal{S}_{j+1}^p)$, so the inequality holds for this domain.

Overall, as $p(\mathcal{S})$ is not a continuous function, we can successfully prove that continuous $p(\mathcal{S}) \le p[vol(\mathcal{S})]$. $\square$

**Lemma 27.** *For any edge set $\mathcal{A} \in \mathcal{V} \times \mathcal{V}$, $p(\mathcal{A}) \leq p[|\mathcal{A}|]$.*

*Proof.* Given $B_j = \{(u,v) \in E | u \in \mathcal{S}_j^P\}$, and $f(u,v) = \phi(u) \cdot p(u,v)$, with $g(u) = \frac{p(u)}{\phi(u)}$ if $(u,v) \in B_j$, we are going to prove from contradiction:

If

$$\sum_{(u,v)\in A} g(u) \cdot f(u,v) > \sum_{(u,v)\in B_j} g(u) \cdot f(u,v) \tag{25}$$

Then, we have

$$\frac{\sum_{(u,v)\in A} g(u) \cdot f(u,v)}{\sum_{(u,v)\in A} f(u,v)} > \frac{\sum_{(u,v)\in B_j} g(u) \cdot f(u,v)}{\sum_{(u,v)\in B_j} f(u,v)} \geq g(v_j) \tag{26}$$

Then, there exists $(u_1, v_1) \in A$ with $u_1 \in S_{j-1}$ such that letting $A^1 = A \setminus \{(u_1, v_1)\}$, $B_j^1 = B_j \setminus \{(u_1, v_1)\}$, we have:

$$\sum_{(u,v)\in A^1} f(u,v) = \sum_{(u,v)\in B_j^1} f(u,v) \tag{27}$$

$$\sum_{(u,v)\in A^1} g(u) \cdot f(u,v) > \sum_{(u,v)\in B_j^1} g(u) \cdot f(u,v) \tag{28}$$

$$\frac{\sum_{(u,v)\in A^1} g(u) \cdot f(u,v)}{\sum_{(u,v)\in A^1} f(u,v)} > \frac{\sum_{(u,v)\in B_j^1} g(u) \cdot f(u,v)}{\sum_{(u,v)\in B_j^1} f(u,v)} \geq g(v_j) \tag{29}$$

Therefore, similarly, for some edge $(u_2, w_2) \in A^1$, where $u \in S_{j-1}^p$, letting $A^2 = A^1 \setminus \{(u,w)\}$; $B_j^2 = B_j^1 \setminus \{(u,w)\}$, It holds that

$$\sum_{(u,v)\in A^2} f(u,v) = \sum_{(u,v)\in B_j^2} f(u,v) \tag{30}$$

and

$$\sum_{(u,v)\in A^2} g(u) \cdot f(u,v) > \sum_{(u,v)\in B_j^2} g(u) \cdot f(u,v) \tag{31}$$

Consequently, we keep this iteration until for some $k$, $A^k = \phi$ or $B_j^k = \phi$, as for empty set, and the definition that $f(u,v) \geq 0$

$$\sum_{(u,v)\in A^k} f(u,v) = \sum_{(u,v)\in B_j^k} f(u,v) = 0 \tag{32}$$

Then, we can get

$$\sum_{(u,v)\in A^1} k \cdot g(u) \cdot f(u,v) = \sum_{(u,v)\in B_j^1} k \cdot g(u) \cdot f(u,v) = 0 \tag{33}$$

which implies

$$\sum_{(u,v)\in A} g(u) \cdot f(u,v) = \sum_{(u,v)\in B_j} g(u) \cdot f(u,v) \tag{34}$$

which is contradictory to the assumption (Equation 25). $\square$

## 4.2 Upper Bound of the Disagreement

In this part, we aim to set up an upper bound of $p[k] - k, \forall k \in [0,1]$. For any vertex set, The following definition describes the edges going into it and the edges going outside of it.

**Definition 28.** *For any vertex set $\mathcal{S} \in \mathcal{V}$, we denote $in(\mathcal{S}) = \{(u,v)|u \in \mathcal{V}, v \in \mathcal{S}\}$; $out(S) = \{(u,v)|u \in \mathcal{S}, v \in \mathcal{V}\}$.*

**Lemma 29.** *For any distribution $p$ and any vertex set $\mathcal{S} \subseteq \mathcal{V}$, recall definition 24,*

$$(pM)(S) = \frac{1}{2}p(in(\mathcal{S})) + \frac{1}{2}p(out(\mathcal{S})) = \frac{1}{2}p(in(\mathcal{S}) \cup out(\mathcal{S})) + \frac{1}{2}p(in(\mathcal{S}) \cap p(out(\mathcal{S})) \quad (35)$$

*Proof.* For any node $u \in \mathcal{V}$, we have $\phi(u) = \sum_{v \in \mathcal{V}} \phi(v) P_{v,u}$

$$(pM)(\{u\}) = (\frac{1}{2}pI + \frac{1}{2}pP)(\{u\}) = \frac{1}{2}p(u) + \frac{1}{2}pP(u) = \frac{1}{2}p(u) + \frac{1}{2}\sum_{v \in \mathcal{V}} p(v) P_{v,u} \quad (36)$$

$$p(out(\{u\})) = \sum_{v \in \mathcal{V}} \phi(u) p(u,v) = \sum_{v \in \mathcal{V}} \phi(u) \frac{p(u)}{\phi(u)} P_{u,v} = \sum_{v \in \mathcal{V}} p(u) P_{u,v} = p(u) \quad (37)$$

$$p(in(\{u\})) = \sum_{v \in \mathcal{V}} \phi(v) p(v,u) = \sum_{v \in \mathcal{V}} \phi(v) \frac{p(v)}{\phi(v)} P_{v,u} = \sum_{v \in \mathcal{V}} p(v) P_{v,u} = pP(u) \quad (38)$$

Therefore, $(pM)(\{u\}) = \frac{1}{2}p(in(\{u\})) + \frac{1}{2}p(out(\{u\}))$ and

$$\begin{aligned}
(pM)(\mathcal{S}) &= \sum_{u \in \mathcal{S}} (pM)(\{u\}) = \sum_{u \in \mathcal{S}} \frac{1}{2}p(in(\{u\})) + \frac{1}{2}p(out(\{u\})) \\
&= \frac{1}{2}\sum_{u \in \mathcal{S}} p(in(\{u\})) + \frac{1}{2}\sum_{u \in \mathcal{S}} p(out(\{u\})) \\
&= \frac{1}{2}p(in(\mathcal{S})) + \frac{1}{2}p(out(\mathcal{S})) \\
&= \frac{1}{2}p(in(\mathcal{S}) \cup out(\mathcal{S})) + \frac{1}{2}p(in(\mathcal{S}) \cap p(out(\mathcal{S}))
\end{aligned} \quad (39)$$

The last equation holds because for any two edge sets $\mathcal{A}$, $\mathcal{B}$, let $k(u,v) = p(u)P_{u,v}$, then,

$$p(\mathcal{A}) + p(\mathcal{B}) = \sum_{(u,v) \in \mathcal{A}} k(u,v) + \sum_{(u,v) \in \mathcal{B}} k(u,v) = \sum_{(u,v) \in \mathcal{A} \cup \mathcal{B}} k(u,v) + \sum_{(u,v) \in \mathcal{A} \cap \mathcal{B}} k(u,v) = p(\mathcal{A} \cup \mathcal{B}) + p(\mathcal{A} \cap \mathcal{B}) \quad (40)$$

$\square$

**Lemma 30.** *Recall the definition of $|\mathcal{A}|$ from definition 25. For any two sets of node pairs $\mathcal{A}, \mathcal{B}$,*

$$|\mathcal{A}| + |\mathcal{B}| = |\mathcal{A} \cup \mathcal{B}| + |\mathcal{A} \cap \mathcal{B}| \quad (41)$$

*Proof.*

$$\begin{aligned}
|\mathcal{A}| + |\mathcal{B}| &= \sum_{(u,v) \in \mathcal{A}} \phi(u) P_{u,v} + \sum_{(u,v) \in \mathcal{B}} \phi(u) P_{u,v} \\
&= \sum_{(u,v) \in \mathcal{A} \cup \mathcal{B}} \phi(u) P_{u,v} + \sum_{(u,v) \in \mathcal{A} \cap \mathcal{B}} \phi(u) P_{u,v} \\
&= |\mathcal{A} \cup \mathcal{B}| + |\mathcal{A} \cap \mathcal{B}|
\end{aligned} \quad (42)$$

$$\square$$

**Lemma 31.** *Recall the definition of $|\mathcal{A}|$ from definition 25, $in(\mathcal{S})$ and $out(\mathcal{S})$ from definition 28, and $vol(\mathcal{S})$ from definition 12. For any vertex set $\mathcal{S} \subseteq \mathcal{V}$,*

$$|in(\mathcal{S})| + |out(\mathcal{S})| = 2vol(\mathcal{S}) \tag{43}$$

*Proof.*

$$
\begin{aligned}
|in(\mathcal{S})| + |out(\mathcal{B})| &= |\{(u,v)|u \in \mathcal{V}, v \in \mathcal{S}\}| + |\{(u,v)|u \in \mathcal{S}, v \in \mathcal{V}\}| \\
&= \sum_{v \in \mathcal{V}, u \in \mathcal{S}} \phi(v)P_{v,u} + \sum_{u \in \mathcal{S}, v \in \mathcal{V}} \phi(u)P_{u,v} \\
&= \sum_{u \in \mathcal{S}} \sum_{v \in \mathcal{V}} \phi(v)P_{v,u} + \sum_{u \in \mathcal{S}} \sum_{v \in \mathcal{V}} \phi(u)P_{u,v} \\
&= \sum_{u \in \mathcal{S}} \phi(u) + \sum_{u \in \mathcal{S}} \phi(u) \sum_{v \in \mathcal{V}} P_{u,v} \\
&= \sum_{u \in \mathcal{S}} \phi(u) + \sum_{u \in \mathcal{S}} \phi(u) \sum_{v \in \mathcal{V}} P_{u,v} \\
&= \sum_{u \in \mathcal{S}} \phi(u) + \sum_{u \in \mathcal{S}} \phi(u) \\
&= vol(S) + vol(S) \\
&= 2vol(S)
\end{aligned}
\tag{44}
$$

$$\square$$

In fact, from the above proof, we have a stronger conclusion that $|in(\mathcal{S})| = |out(\mathcal{S})| = vol(\mathcal{S})$.

**Lemma 32.** *Recall the definition of $|\mathcal{A}|$ from definition 25, and $|\partial\mathcal{S}|$ from definition 11. For any vertex set $\mathcal{S} \subseteq \mathcal{V}$,*

$$|in(\mathcal{S}) \cup out(\mathcal{S})| - |in(\mathcal{S}) \cap out(\mathcal{S})| = 2|\partial\mathcal{S}| \tag{45}$$

*Proof.*

$$
\begin{aligned}
|in(\mathcal{S}) \cup out(\mathcal{B})| + |in(\mathcal{B}) \cap out(\mathcal{B})| &= \sum_{(u \in \mathcal{S}, v \in \mathcal{V}) \vee (u \in \mathcal{V}, v \in \mathcal{S})} \phi(u)P_{u,v} - \sum_{(u \in \mathcal{S}, v \in \mathcal{V}) \wedge (u \in \mathcal{V}, v \in \mathcal{S})} \phi(u)P_{u,v} \\
&= \left( \sum_{u \in \bar{\mathcal{S}}, v \in \mathcal{S}} \phi(u)P_{u,v} + \sum_{u \in \mathcal{S}, v \in \bar{\mathcal{S}}} \phi(u)P_{u,v} + \sum_{u \in \mathcal{S}, v \in \mathcal{S}} \phi(u)P_{u,v} \right) - \sum_{u \in \mathcal{S}, v \in \mathcal{S}} \phi(u)P_{u,v} \\
&= \sum_{u \in \bar{\mathcal{S}}, v \in \mathcal{S}} \phi(u)P_{u,v} + \sum_{u \in \mathcal{S}, v \in \bar{\mathcal{S}}} \phi(u)P_{u,v} \\
&= |\partial\mathcal{S}| + |\partial\mathcal{S}| \\
&= 2|\partial\mathcal{S}|
\end{aligned}
\tag{46}
$$

$$\square$$

**Lemma 33.** *For any vertex set $\mathcal{S} \subseteq \mathcal{V}$,*

$$
\begin{aligned}
|in(\mathcal{S}) \cup out(\mathcal{S})| &= vol(\mathcal{S}) + |\partial\mathcal{S}| \\
|in(\mathcal{S}) \cap out(\mathcal{S})| &= vol(\mathcal{S}) - |\partial\mathcal{S}|
\end{aligned}
\tag{47}
$$

*Proof.* From Lemma 30 and Lemma 31,

$$|in(\mathcal{S}) \cup out(\mathcal{S})| + |in(\mathcal{S}) \cap out(\mathcal{S})| = |in(\mathcal{S})| + |out(\mathcal{S})| = 2vol(\mathcal{S}) \tag{48}$$

From Lemma 32,

$$|in(\mathcal{S}) \cup out(\mathcal{S})| - |in(\mathcal{S}) \cap out(\mathcal{S})| = 2|\partial\mathcal{S}| \tag{49}$$

The above two equations give

$$\begin{aligned} |in(\mathcal{S}) \cup out(\mathcal{S})| &= vol(\mathcal{S}) + |\partial\mathcal{S}| \\ |in(\mathcal{S}) \cap out(\mathcal{S})| &= vol(\mathcal{S}) - |\partial\mathcal{S}| \end{aligned} \tag{50}$$

$\square$

**Lemma 34.** *If $p = pr(\alpha, s)$ is a lazy PageRank vector, then $\forall \mathcal{S} \in \mathcal{V}$,*

$$p(\mathcal{S}) = \alpha s(\mathcal{S}) + (1 - \alpha)\frac{1}{2}(p(in(\mathcal{S}) \cup out(\mathcal{S})) + p(in(\mathcal{S}) \cap out(\mathcal{S}))) \tag{51}$$

*Furthermore, recall the definition of $p[\cdot]$ from definition 22, for each $j \in [1, N_p]$,*

$$p[vol(\mathcal{S}_j^p)] \le \alpha s[vol(\mathcal{S}_j^p)] + (1 - \alpha)\frac{1}{2}(p[vol(\mathcal{S}_j^p) + |\partial\mathcal{S}_j^p|] + p[vol(\mathcal{S}_j^p) - |\partial\mathcal{S}_j^p|]) \tag{52}$$

*Proof.* $\forall \mathcal{S} \in \mathcal{V}$,

$$\begin{aligned} p(\mathcal{S}) = pr(\alpha, s) &= (\alpha s + (1 - \alpha pM))(\mathcal{S}) \\ &= \alpha s(\mathcal{S}) + ((1 - \alpha)pM)(\mathcal{S}) \\ &\overset{Lemma\ 29}{=} \alpha s(\mathcal{S}) + (1 - \alpha)\frac{1}{2}(p(in(\mathcal{S}) \cup out(\mathcal{S})) + p(in(\mathcal{S}) \cap out(\mathcal{S}))) \end{aligned} \tag{53}$$

Recall from definition 22 that $p[vol(\mathcal{S}_j^p)] = p(\mathcal{S}_j^p)$,

$$\begin{aligned} p[vol(\mathcal{S}_j^p)] &= p(\mathcal{S}_j^p) \\ &= \alpha s(\mathcal{S}_j^p) + (1 - \alpha)\frac{1}{2}(p(in(\mathcal{S}_j^p) \cup out(\mathcal{S}_j^p)) + p(in(\mathcal{S}_j^p) \cap out(\mathcal{S}_j^p))) \\ &\overset{Lemma\ 27}{\le} \alpha s(\mathcal{S}_j^p) + (1 - \alpha)\frac{1}{2}(p[|in(\mathcal{S}_j^p) \cup out(\mathcal{S}_j^p)|] + p[|in(\mathcal{S}_j^p) \cap out(\mathcal{S}_j^p)|]) \\ &\overset{Lemma\ 33}{=} \alpha s(\mathcal{S}_j^p) + (1 - \alpha)\frac{1}{2}(p[vol(\mathcal{S}_j^p) + |\partial S_j^p|] + p[vol(\mathcal{S}_j^p) - |\partial S_j^p|]) \\ &\overset{Lemma\ 26}{\le} \alpha s[vol(\mathcal{S}_j^p)] + (1 - \alpha)\frac{1}{2}(p[vol(\mathcal{S}_j^p) + |\partial S_j^p|] + p[vol(\mathcal{S}_j^p) - |\partial S_j^p|]) \end{aligned} \tag{54}$$

$\square$

**Theorem 35.** *Let $\sigma$ $\gamma$ and $\theta$ be any constants such that $\sigma \in [0, 1], \gamma \in [0, 1], \theta \in [0, \frac{1}{2}]$. Let $p = pr(\alpha, s)$ be a lazy PageRank vector, then **either** the following bound holds for any integer $t$ and any $k \in [\theta, 1 - \theta] \cup \{0, 1\}$,*

$$p[k] - k \le \gamma + \alpha t + \sqrt{\frac{1}{\theta}}\sqrt{\min(k, 1 - k)}(1 - \frac{\sigma^2}{8})^t \tag{55}$$

*or there exists a sweep cut $\mathcal{S}_j^p, j \in [1, N_p]$, with the following properties,*

Property 1. $\Phi(\mathcal{S}_j^p) < \sigma$

Property 2. $\exists t \in \mathbb{N}_+, p(\mathcal{S}_j^p) - vol(\mathcal{S}_j^p) > \gamma + \alpha t + \sqrt{\frac{1}{\theta}}\sqrt{\min(vol(\mathcal{S}_j^p), 1 - vol(\mathcal{S}_j^p))}(1 - \frac{\sigma^2}{8})^t$ (56)

*Proof.* Let $k_j = vol(\mathcal{S}_j^p), \bar{k}_j = \min(k_j, 1 - k_j)$. Let

$$f_t(k) = \gamma + \alpha t + \sqrt{\frac{1}{\theta}}\sqrt{\min(k, 1 - k)}(1 - \frac{\sigma^2}{8})^t \tag{57}$$

For $k = 0$ or $k = 1$, $p[k] - k = 0 \le f_t(k)$, and the "either" side holds.

If there is no sweep cut satisfying the two properties at the "or" side, we aim to prove the "either" side holds. In other words, we aim to prove that, for all $t \ge 0$ and $k \in [\theta, 1 - \theta]$,

$$p[k] - k \le f_t(k) \tag{58}$$

We prove this by induction. When $t = 0$, $f_t(k) = \gamma + \sqrt{\frac{1}{\theta}}\sqrt{\min(k, 1 - k)} \ge \sqrt{\frac{1}{\theta}}\sqrt{\min(k, 1 - k)}$,

$$p[k] - k \le p[k] \le 1 = \sqrt{\frac{1}{\theta}}\sqrt{\theta} \overset{k \in [\theta, 1-\theta]}{\le} \sqrt{\frac{1}{\theta}}\sqrt{\min(k, 1 - k)} \tag{59}$$

**Assume for the induction that**, for $t$, the inequality 58: $p[k] - k \le f_t(k)$ holds. we aim to prove that the inequality 58 also holds for $t + 1$, i.e., $p[k] - k \le f_{t+1}(k)$.

Note the fact that the sum of two concave functions is still concave. With respect to $k$, since $\sqrt{\min(k, 1 - k)}$ is concave, we have $f_{t+1}(k)$ is concave, and furthermore $f_{t+1}(k) + k$ is concave. To show $p[k] - k \le f_{t+1}(k)$, it suffices to prove

$$p[k_j] \le f_{t+1}(k_j) + k_j \tag{60}$$

Then, since $f_{t+1}(k) + k$ is concave and $p[k]$ is piecewise linear, $\forall k_j < k < k_{j+1}$ we have $p[k] \le f_{t+1}(k) + k$.

Therefore, to finish the induction, we only need to prove the inequality 60. Since there is no sweep cut satisfying the two properties, $\forall j \in [1, |Supp(p)|]$ (recall definition 19), $\mathcal{S}_j^p$ does not satisfy Property 1 or Property 2.

**If** $\mathcal{S}_j^p$ does not satisfy property 2, then $\forall t \in \mathbb{N}_+$,

$$p(\mathcal{S}_j^p) - vol(\mathcal{S}_j^p) \le \gamma + \alpha t + \sqrt{\frac{1}{\theta}}\sqrt{\min(vol(\mathcal{S}_j^p), 1 - vol(\mathcal{S}_j^p))}(1 - \frac{\sigma^2}{8})^t$$
$$\iff p[vol(\mathcal{S}_j^p)] - vol(\mathcal{S}_j^p) \le f_t(vol(\mathcal{S}_j^p)) \tag{61}$$
$$\iff p[k_j] - k_j \le f_t(k_j)$$

**Else**, $\mathcal{S}_j^p$ does not satisfy property 1. Then, $\Phi(\mathcal{S}_j^p) \ge \sigma$. Since $p[k]$ is a monotonically non-decreasing piecewise linear concave function,

$$\text{when } b \ge c, \ p[a - b] + p[a + b] \le p[a - c] + p[a + c] \tag{62}$$

$$
\begin{aligned}
p[k_j] &\overset{k_j=vol(\mathcal{S}_j^p)}{=} p[vol(\mathcal{S}_j^p)] \\
&\overset{Lemma\ 34}{\leq} \alpha s[vol(\mathcal{S}_j^p)] + (1-\alpha)\tfrac{1}{2}(p[vol(\mathcal{S}_j^p)+|\partial\mathcal{S}_j^p|] + p[vol(\mathcal{S}_j^p)-|\partial\mathcal{S}_j^p|]) \\
&\overset{switch\ terms}{=} \alpha s[vol(\mathcal{S}_j^p)] + (1-\alpha)\tfrac{1}{2}(p[vol(\mathcal{S}_j^p)-|\partial\mathcal{S}_j^p|] + p[vol(\mathcal{S}_j^p)+|\partial\mathcal{S}_j^p|]) \\
&\overset{Definition\ 11}{=} \alpha s[vol(\mathcal{S}_j^p)] + (1-\alpha)\tfrac{1}{2}(p[vol(\mathcal{S}_j^p)-\Phi(\mathcal{S}_j^p)\bar{k}_j] + p[vol(\mathcal{S}_j^p)+\Phi(\mathcal{S}_j^p)\bar{k}_j]) \\
&\overset{Equation\ 62}{\leq} \alpha s[vol(\mathcal{S}_j^p)] + (1-\alpha)\tfrac{1}{2}(p[vol(\mathcal{S}_j^p)-\sigma\bar{k}_j] + p[vol(\mathcal{S}_j^p)+\sigma\bar{k}_j]) \\
&\overset{\alpha\in[0,1],s[k]\leq 1}{\leq} \alpha + \tfrac{1}{2}(p[vol(\mathcal{S}_j^p)-\sigma\bar{k}_j] + p[vol(\mathcal{S}_j^p)+\sigma\bar{k}_j]) \\
&\overset{k_j=vol(\mathcal{S}_j^p)}{=} \alpha + \tfrac{1}{2}(p[k_j-\sigma\bar{k}_j] + p[k_j+\sigma\bar{k}_j]) \\
&\overset{induction\ hypothesis}{\leq} \alpha + \tfrac{1}{2}(f_t(k_j-\sigma\bar{k}_j) + (k_j-\sigma\bar{k}_j) + f_t(k_j+\sigma\bar{k}_j) + (k_j+\sigma\bar{k}_j)) \\
&= \alpha + k_j + \tfrac{1}{2}(f_t(k_j-\sigma\bar{k}_j) + f_t(k_j+\sigma\bar{k}_j))
\end{aligned}
\tag{63}
$$

Therefore,

$$
\begin{aligned}
p[k_j] - k_j &\leq \alpha + \frac{1}{2}(f_t(k_j-\sigma\bar{k}_j) + f_t(k_j+\sigma\bar{k}_j)) \\
&= \alpha + \gamma + \alpha t + \frac{1}{2}\sqrt{\frac{1}{\theta}}(1-\frac{\sigma^2}{8})^t(\sqrt{\min(k_j-\sigma\bar{k}_j, 1-k_j+\sigma\bar{k}_j)} + \sqrt{\min(k_j+\sigma\bar{k}_j, 1-k_j-\sigma\bar{k}_j)})
\end{aligned}
\tag{64}
$$

**1°** When $k_j \in [0, \frac{1}{2}], \bar{k}_j = k_j$.

$$
\begin{aligned}
&\sqrt{\min(k_j-\sigma\bar{k}_j, 1-k_j+\sigma\bar{k}_j)} + \sqrt{\min(k_j+\sigma\bar{k}_j, 1-k_j-\sigma\bar{k}_j)} \\
&= \sqrt{\min(\bar{k}_j-\sigma\bar{k}_j, 1-k_j+\sigma\bar{k}_j)} + \sqrt{\min(\bar{k}_j+\sigma\bar{k}_j, 1-k_j-\sigma\bar{k}_j)} \\
&\leq \sqrt{\bar{k}_j-\sigma\bar{k}_j} + \sqrt{\bar{k}_j+\sigma\bar{k}_j}
\end{aligned}
\tag{65}
$$

**2°** When $k_j \in [\frac{1}{2}, 1], \bar{k}_j = 1 - k_j$.

$$
\begin{aligned}
&\sqrt{\min(k_j-\sigma\bar{k}_j, 1-k_j+\sigma\bar{k}_j)} + \sqrt{\min(k_j+\sigma\bar{k}_j, 1-k_j-\sigma\bar{k}_j)} \\
&= \sqrt{\min(1-\bar{k}_j-\sigma\bar{k}_j, \bar{k}_j+\sigma\bar{k}_j)} + \sqrt{\min(1-\bar{k}_j+\sigma\bar{k}_j, \bar{k}_j-\sigma\bar{k}_j)} \\
&\leq \sqrt{\bar{k}_j+\sigma\bar{k}_j} + \sqrt{\bar{k}_j-\sigma\bar{k}_j} \\
&= \sqrt{\bar{k}_j-\sigma\bar{k}_j} + \sqrt{\bar{k}_j+\sigma\bar{k}_j}
\end{aligned}
\tag{66}
$$

Therefore, $\forall k_j \in [0,1], \sqrt{\min(k_j-\sigma\bar{k}_j, 1-k_j+\sigma\bar{k}_j)} + \sqrt{\min(k_j+\sigma\bar{k}_j, 1-k_j-\sigma\bar{k}_j)} \leq \sqrt{\bar{k}_j-\sigma\bar{k}_j} + \sqrt{\bar{k}_j+\sigma\bar{k}_j}$. Continue with Equation 64,

$$p[k_j] - k_j \leq \alpha + \gamma + \alpha t + \frac{1}{2}(1 - \frac{\sigma^2}{8})^t \sqrt{\frac{1}{\theta}}(\sqrt{\min(k_j - \sigma\bar{k}_j, 1 - k_j + \sigma\bar{k}_j)} + \sqrt{\min(k_j + \sigma\bar{k}_j, 1 - k_j - \sigma\bar{k}_j)})$$

$$\leq \alpha + \gamma + \alpha t + \frac{1}{2}(1 - \frac{\sigma^2}{8})^t \sqrt{\frac{1}{\theta}}(\sqrt{\bar{k}_j - \sigma\bar{k}_j} + \sqrt{\bar{k}_j + \sigma\bar{k}_j})$$

$$(67)$$

Consider the Taylor series

$$\sqrt{1 + x} = (1 + x)^{\frac{1}{2}} = \sum_{n=0}^{\infty} \binom{\frac{1}{2}}{n} \cdot x^n$$

$$\sqrt{1 - x} = (1 + (-x))^{\frac{1}{2}} = \sum_{n=0}^{\infty} \binom{\frac{1}{2}}{n} \cdot (-x)^n$$

$$(68)$$

$$\sqrt{1 - x} + \sqrt{1 + x} = \sum_{n=0}^{\infty} \binom{\frac{1}{2}}{n} \cdot (-x)^n + \sum_{n=0}^{\infty} \binom{\frac{1}{2}}{n} \cdot x^n = 2\sum_{m=0}^{\infty} \binom{\frac{1}{2}}{2m} \cdot x^{2m}$$

$$\stackrel{\forall m \geq 2, \binom{\frac{1}{2}}{2m} \leq 0, x^{2m} \geq 0}{\leq} 2(\binom{\frac{1}{2}}{0} \cdot x^0 + \binom{\frac{1}{2}}{2} \cdot x^2)$$

$$= 2(1 + \frac{\frac{1}{2} \cdot (-\frac{1}{2})}{2} \cdot x^2)$$

$$= 2(1 - \frac{x^2}{8})$$

$$(69)$$

Continue with Equation 67

$$p[k_j] - k_j \leq \alpha + \gamma + \alpha t + \frac{1}{2}(1 - \frac{\sigma^2}{8})^t \sqrt{\frac{1}{\theta}}(\sqrt{\bar{k}_j - \sigma\bar{k}_j} + \sqrt{\bar{k}_j + \sigma\bar{k}_j})$$

$$= \gamma + \alpha(t + 1) + \frac{1}{2}(1 - \frac{\sigma^2}{8})^t \sqrt{\frac{1}{\theta}}\sqrt{\bar{k}_j}(\sqrt{1 - \sigma} + \sqrt{1 + \sigma})$$

$$\stackrel{Equation\ 67}{\leq} \gamma + \alpha(t + 1) + \frac{1}{2}(1 - \frac{\sigma^2}{8})^t \sqrt{\frac{1}{\theta}}\sqrt{\bar{k}_j} \cdot 2(1 - \frac{\sigma^2}{8})$$

$$= \gamma + \alpha(t + 1) + (1 - \frac{\sigma^2}{8})^{t+1} \sqrt{\frac{1}{\theta}}\sqrt{\bar{k}_j}.$$

$$= \gamma + \alpha(t + 1) + \sqrt{\frac{1}{\theta}}\sqrt{\min(k_j, 1 - k_j)}(1 - \frac{\sigma^2}{8})^{t+1}$$

$$= f_{t+1}(k_j)$$

$$(70)$$

Combining equations 61 and 70, no matter which property the sweep cut does not satisfy, we always have $p[k_j] - k_j \leq f_{t+1}(k_j)$, which completes the induction step. $\qquad\square$

Though Theorem 35 describes an "either or" relation, we can obtain an upper bound by setting $\sigma = \Phi(pr(\alpha, s)) = \min_{j \in [1, N_p]} \Phi(\mathcal{S}_j^{pr(\alpha,s)}) = \min_{j \in [1, N_p]} \Phi(\mathcal{S}_j^p)$. Then, the "or" case cannot hold. Therefore the "either" case holds and we obtain an upper bound of the disagreement $p[k] - k$. Moreover, this upper bound is associated with $\sigma = \Phi(pr(\alpha, s))$.

### 4.3 Solve the Conductance Given Lower Bound

From Theorem 35, since the right-hand side on the "either" side decreases when $\sigma$ increases, if $p[k] - k$ has another lower bound, then we can further upper-bound $\sigma = \Phi(pr(\alpha, s))$.

**Theorem 36.** *For a lazy PageRank vector $pr(\alpha, s)$ and any constant $z \in [0, \frac{1}{e}]$, if there exists a vertex set $\mathcal{S} \subseteq \mathcal{V}$ that satisfies*

$$\delta = pr(\alpha, s)(\mathcal{S}) - vol(\mathcal{S}) > z \tag{71}$$

*then,*

$$\Phi(pr(\alpha, s)) < \sqrt{\frac{9\alpha \ln \frac{1}{z}}{\delta - z}} \tag{72}$$

*Proof.* Recall from Theorem 15 and Definition 20, we have $\Phi(pr(\alpha, s)) \in [0, 1]$. In Theorem 35, let $\sigma = \Phi(pr(\alpha, s)), \gamma = 0, p = pr(\alpha, s)$ is a lazy PageRank vector. Then, for the constant $\theta = \min(vol(\mathcal{S}), 1 - vol(\mathcal{S})) \in [0, \frac{1}{2}]$, **either** the following bound holds for any integer $t$ and any $k \in [\theta, 1 - \theta] \cup \{0, 1\}$

$$p[k] - k \le \alpha t + \sqrt{\frac{1}{\theta}} \sqrt{\min(k, 1 - k)} (1 - \frac{\sigma^2}{8})^t \tag{73}$$

**or** there exist a sweep cut $\mathcal{S}_j^p, j \in [1, N_p]$, with the following properties,

$$\begin{aligned} &\text{Property 1. } \Phi(\mathcal{S}_j^p) < \sigma \\ &\text{Property 2. } \exists t \in \mathbb{N}_+, p(\mathcal{S}_j^p) - vol(\mathcal{S}_j^p) > \alpha t + \sqrt{\frac{1}{\theta}} \sqrt{\min(vol(\mathcal{S}_j^p), 1 - vol(\mathcal{S}_j^p))} (1 - \frac{\sigma^2}{8})^t \end{aligned} \tag{74}$$

As we set $\sigma = \Phi(pr(\alpha, s)) = \min_{j \in [1, N_p]} \Phi(\mathcal{S}_j^{pr(\alpha, s)}) = \min_{j \in [1, N_p]} \Phi(\mathcal{S}_j^p)$, the "or" case cannot hold. Therefore, the "either" case holds and $\forall t \in \mathbb{N}_+$,

$$\begin{aligned} \delta = pr(\alpha, s)(\mathcal{S}) - vol(\mathcal{S}) &\overset{Lemma\ 26}{\le} pr(\alpha, s)[vol(\mathcal{S})] - vol(\mathcal{S}) \\ &\overset{Equation\ 73}{\le} \alpha t + \sqrt{\frac{1}{\theta}} \sqrt{\min(vol(\mathcal{S}), 1 - vol(\mathcal{S}))} (1 - \frac{\sigma^2}{8})^t \\ &= \alpha t + \sqrt{\frac{1}{\min(vol(\mathcal{S}), 1 - vol(\mathcal{S}))}} \sqrt{\min(vol(\mathcal{S}), 1 - vol(\mathcal{S}))} (1 - \frac{\sigma^2}{8})^t \\ &= \alpha t + (1 - \frac{\sigma^2}{8})^t \end{aligned} \tag{75}$$

$\forall z \le \frac{1}{e}, \ln \frac{1}{z} \ge 1$, let $t_z = \frac{8}{\sigma^2} \ln \frac{1}{z}, t = \lceil t_z \rceil \ge t_z$, then,

$$\delta \overset{Equation\ 75}{\leq} \alpha t + (1 - \frac{\sigma^2}{8})^t$$

$$= \alpha \lceil \frac{8}{\sigma^2} \ln \frac{1}{z} \rceil + (1 - \frac{\sigma^2}{8})^{\lceil \frac{8}{\sigma^2} \ln \frac{1}{z} \rceil}$$

$$\leq \alpha \lceil \frac{8}{\sigma^2} \ln \frac{1}{z} \rceil + (1 - \frac{\sigma^2}{8})^{\frac{8}{\sigma^2} \ln \frac{1}{z}}$$

$$\leq \alpha \lceil \frac{8}{\sigma^2} \ln \frac{1}{z} \rceil + (\frac{1}{e})^{\ln \frac{1}{z}} \qquad (76)$$

$$\leq \alpha \lceil \frac{8}{\sigma^2} \ln \frac{1}{z} \rceil + z$$

$$= \alpha (\frac{8}{\sigma^2} \ln \frac{1}{z} + 1) + z$$

$$< \alpha (\frac{9}{\sigma^2} \ln \frac{1}{z}) + z$$

Therefore, if $\delta > z$,

$$\sigma \leq \sqrt{\frac{9\alpha \ln \frac{1}{z}}{\delta - z}} \qquad (77)$$

$\square$

## 4.4 Lower Bound of the Disagreement

In this part, we find a $z$ value in Theorem 36 to bound $\Phi(pr(\alpha, s))$ in the next subsection.

**Lemma 37.** *Recall the definition of $p[\cdot]$ from definition 22. For any starting distribution $s$ and any $k \in [0, 1]$,*

$$pr(\alpha, s)[k] \leq s[k] \qquad (78)$$

*Proof.* Let $p = pr(\alpha, s)$. From Lemma 34, $\forall j \in [1, N_p]$,

$$p[vol(\mathcal{S}_j^p)] \leq \alpha s[vol(\mathcal{S}_j^p)] + (1 - \alpha)\frac{1}{2}(p[vol(\mathcal{S}_j^p) + |\partial \mathcal{S}_j^p|] + p[vol(\mathcal{S}_j^p) - |\partial \mathcal{S}_j^p|])$$

$$\overset{Concavity\ of\ p[k]}{\leq} \alpha s[vol(\mathcal{S}_j^p)] + (1 - \alpha)\frac{1}{2}(p[vol(\mathcal{S}_j^p) + 0] + p[vol(\mathcal{S}_j^p) - 0])$$

$$= \alpha s[vol(\mathcal{S}_j^p)] + (1 - \alpha)(p[vol(\mathcal{S}_j^p)]) \qquad (79)$$

$$\Longleftrightarrow p[vol(\mathcal{S}_j^p)] - (1 - \alpha)(p[vol(\mathcal{S}_j^p)]) \leq \alpha s[vol(\mathcal{S}_j^p)]$$

$$\Longleftrightarrow p[vol(\mathcal{S}_j^p)] \leq s[vol(\mathcal{S}_j^p)]$$

Since $s[k]$ is concave, $p[k]$ is piecewise linear, and the breakpoints $(vol(\mathcal{S}_j^p), p[vol(\mathcal{S}_j^p)])$ are bounded by $(vol(\mathcal{S}_j^p), s[vol(\mathcal{S}_j^p)])$, i.e., for each $k_j = vol(\mathcal{S}_j^p), p[k_j] \leq s[k_j]$, we have

$$p[k] \leq s[k] \iff pr(\alpha, s) \leq s[k] \ \forall k \in [0, 1]. \qquad (80)$$

$\square$

**Definition 38.** *For any vertex set $\mathcal{S} \subseteq \mathcal{V}$, we denote $\partial_{in}(\mathcal{S}) = \{(u, v)|u \in \bar{\mathcal{S}}, v \in \mathcal{S}\}$; $\partial_{out}(S) = \{(u, v)|u \in \mathcal{S}, v \in \bar{\mathcal{S}}\}$. Then, by Definition 25, we have $|\partial_{in}(\mathcal{S})| = \sum_{u \in \bar{\mathcal{S}}, v \in \mathcal{S}} \phi(u)P_{u,v}, |\partial_{out}(\mathcal{S})| = \sum_{u \in \mathcal{S}, v \in \bar{\mathcal{S}}} \phi(u)P_{u,v}$.*

Here, we give another proof for Theorem 10 which shows $|\partial_{in}(\mathcal{S})| = |\partial_{out}(\mathcal{S})| = |\partial \mathcal{S}|$.

**Lemma 39.** *For any vertex set, referring to definition 38, $|\partial_{in}(\mathcal{S})| = |\partial_{out}(\mathcal{S})| = |\partial\mathcal{S}|$.*

*Proof.* Recall from the proof of Lemma 31, we have

$$\sum_{u\in\mathcal{S},v\in\mathcal{V}} \phi(u)P_{u,v} = |out(\mathcal{S})| = vol(\mathcal{S})$$
$$\sum_{u\in\mathcal{V},v\in\mathcal{S}} \phi(u)P_{u,v} = \sum_{u\in\mathcal{S},v\in\mathcal{V}} \phi(v)P_{v,u} = |in(\mathcal{S})| = vol(\mathcal{S}) \tag{81}$$

$$|out(S)| - \sum_{u\in\mathcal{S},v\in\mathcal{S}} \phi(u)P_{u,v} = \sum_{u\in\mathcal{S},v\in\mathcal{V}} \phi(u)P_{u,v} - \sum_{u\in\mathcal{S},v\in\mathcal{S}} \phi(u)P_{u,v} = \sum_{u\in\mathcal{S},v\in\bar{\mathcal{S}}} \phi(u)P_{u,v} = |\partial_{out}(\mathcal{S})|$$
$$|in(S)| - \sum_{u\in\mathcal{S},v\in\mathcal{S}} \phi(u)P_{u,v} = \sum_{u\in\mathcal{V},v\in\mathcal{S}} \phi(u)P_{u,v} - \sum_{u\in\mathcal{S},v\in\mathcal{S}} \phi(u)P_{u,v} = \sum_{u\in\bar{\mathcal{S}},v\in\mathcal{S}} \phi(u)P_{u,v} = |\partial_{in}(\mathcal{S})| \tag{82}$$

Therefore, $|\partial_{out}(\mathcal{S})| = vol(\mathcal{S}) - |out(\mathcal{S}) \cap in(\mathcal{S})| = |\partial_{in}(\mathcal{S})|$.

$\square$

**Definition 40.** *For any vertex set $\mathcal{C} \subseteq \mathcal{V}$, we define its PageRank Starting Distribution*

$$\psi_\mathcal{C}(v) = \begin{cases} \frac{\phi(v)}{vol(\mathcal{C})} & , \ v \in \mathcal{C} \\ 0 & , \ otherwise \end{cases} \tag{83}$$

**Theorem 41.** *For any vertex set $\mathcal{C} \subseteq \mathcal{V}$, recall the definition of lazy PageRank vector in Definition 16 and Definition 9,*

$$pr(\alpha, \psi_\mathcal{C})(\bar{\mathcal{C}}) \le \frac{\Phi(\mathcal{C})}{2\alpha} \tag{84}$$

*Proof.* From the proof of Lemma 39, we have $|\partial_{out}(\mathcal{S})| \le out(\mathcal{S}) = vol(\mathcal{S}), |\partial_{in}(\mathcal{S})| \le in(\mathcal{S}) = vol(\mathcal{S})$. By definition 40, $\forall v \in \mathcal{C}, \frac{\psi_\mathcal{C}(v)}{\phi(v)} = vol(\mathcal{V}); \forall v \notin \mathcal{C}, \frac{\psi_\mathcal{C}(v)}{\phi(v)} = 0$. Therefore,

$$\psi_\mathcal{C}[k] = \begin{cases} \frac{k}{vol(\mathcal{C})} & , \ k \in [0, vol(\mathcal{C})] \\ 1 & , \ k \in [vol(\mathcal{C}), 1] \end{cases} \tag{85}$$

By Lemma 37,

$$pr(\alpha, \psi_\mathcal{C})[|\partial_{out}(\mathcal{S})|] \le \psi_\mathcal{C}[|\partial_{out}(\mathcal{S})|] = \frac{|\partial_{out}(\mathcal{S})|}{vol(\mathcal{C})}$$
$$pr(\alpha, \psi_\mathcal{C})[|\partial_{in}(\mathcal{S})|] \le \psi_\mathcal{C}[|\partial_{in}(\mathcal{S})|] = \frac{|\partial_{in}(\mathcal{S})|}{vol(\mathcal{C})} \tag{86}$$

For one-step random walk and any distribution $s$, $sP(\bar{\mathcal{C}}) - s(\bar{\mathcal{C}})$ is the amount of probability pushed to $\bar{\mathcal{C}}$ subtracted by the amount of probability pushed from $\bar{\mathcal{C}}$. Therefore,

$$sP(\bar{\mathcal{C}}) - s(\bar{\mathcal{C}}) = \sum_{u\in\mathcal{C},v\in\bar{\mathcal{C}}} s(u)P_{u,v} - \sum_{u\in\bar{\mathcal{C}},v\in\mathcal{C}} s(u)P_{u,v} \le \sum_{u\in\mathcal{C},v\in\bar{\mathcal{C}}} s(u)P_{u,v} \tag{87}$$
$$\stackrel{Definition\,24}{=} s(\partial_{out}(\mathcal{C})) \stackrel{Lemma\,27}{\le} s[|\partial_{out}(\mathcal{C})|]$$

and

$$
\begin{aligned}
pr(\alpha, \psi_{\mathcal{C}})(\bar{\mathcal{C}}) &= (\alpha \psi_{\mathcal{C}} + (1 - \alpha) pr(\alpha, \psi_{\mathcal{C}}) M)(\bar{\mathcal{C}}) \\
&\stackrel{\psi_{\mathcal{C}}(\bar{\mathcal{C}})=0}{=} ((1 - \alpha) pr(\alpha, \psi_{\mathcal{C}}) M)(\bar{\mathcal{C}}) \\
&\stackrel{M=\frac{1}{2}(I+P)}{=} \frac{(1 - \alpha)}{2} pr(\alpha, \psi_{\mathcal{C}})(\bar{\mathcal{C}}) + \frac{(1 - \alpha)}{2} (pr(\alpha, \psi_{\mathcal{C}}) P)(\bar{\mathcal{C}}) \\
&\stackrel{Equation\ 87}{\leq} \frac{(1 - \alpha)}{2} pr(\alpha, \psi_{\mathcal{C}})(\bar{\mathcal{C}}) + \frac{(1 - \alpha)}{2} (pr(\alpha, \psi_{\mathcal{C}})(\bar{\mathcal{C}}) + pr(\alpha, \psi_{\mathcal{C}})[|\partial_{out}(\mathcal{C})|]) \\
&= (1 - \alpha) pr(\alpha, \psi_{\mathcal{C}})(\bar{\mathcal{C}}) + \frac{(1 - \alpha)}{2} pr(\alpha, \psi_{\mathcal{C}})[|\partial_{out}(\mathcal{C})|]
\end{aligned}
\tag{88}
$$

Hence,

$$
\begin{aligned}
\alpha pr(\alpha, \psi_{\mathcal{C}})(\bar{\mathcal{C}}) &\leq \frac{(1 - \alpha)}{2} pr(\alpha, \psi_{\mathcal{C}})[|\partial_{out}(\mathcal{C})|] \\
&\stackrel{Equation\ 86}{\leq} \frac{(1 - \alpha)}{2} \frac{[|\partial_{out}(\mathcal{C})|]}{vol(\mathcal{C})} \\
&\stackrel{Lemma\ 39}{\leq} \frac{(1 - \alpha)}{2} \frac{[|\partial\mathcal{C}|]}{vol(\mathcal{C})} \\
&\leq \frac{(1 - \alpha)}{2} \frac{[|\partial\mathcal{C}|]}{\min(vol(\mathcal{C}), 1 - vol(\mathcal{C}))} \\
&= \frac{1 - \alpha}{2} \Phi(\mathcal{C})
\end{aligned}
\tag{89}
$$

$$
\therefore pr(\alpha, \psi_{\mathcal{C}})(\bar{\mathcal{C}}) \leq \frac{1 - \alpha}{2\alpha} \Phi(\mathcal{C}) \leq \frac{\Phi(\mathcal{C})}{2\alpha}
\tag{90}
$$

Once we have $pr(\alpha, \psi_{\mathcal{C}})(\bar{\mathcal{C}}) \leq \frac{\Phi(\mathcal{C})}{2\alpha}$, we can obtain $pr(\alpha, \psi_{\mathcal{C}})(\mathcal{C}) \geq 1 - \frac{\Phi(\bar{\mathcal{C}})}{2\alpha}$ and finish the proof of the quadratic optimality of the smallest sweep cut conductance.

$\square$

## 4.5 Quadratic Optimality

**Theorem 42.** *For any vertex set $\mathcal{C} \subseteq \mathcal{V}$, sample a subset $\mathcal{D} \subseteq \mathcal{C}$ uniformly randomly. Then, with probability at least $\frac{1}{2}$,*

$$
pr(\alpha, \psi_{\mathcal{D}})(\bar{\mathcal{C}}) \leq pr(\alpha, \psi_{\mathcal{C}})(\bar{\mathcal{C}})
\tag{91}
$$

*Moreover, at least one of the following equations holds.*

$$
pr(\alpha, \psi_{\mathcal{D}})(\bar{\mathcal{C}}) \leq pr(\alpha, \psi_{\mathcal{C}})(\bar{\mathcal{C}})
\tag{92}
$$

$$
pr(\alpha, \psi_{\mathcal{C} \setminus \mathcal{D}})(\bar{\mathcal{C}}) \leq pr(\alpha, \psi_{\mathcal{C}})(\bar{\mathcal{C}})
\tag{93}
$$

*Proof.* By definition 40,

$$
\psi_{\mathcal{C}}(v) = \begin{cases} \frac{\phi(v)}{vol(\mathcal{C})} & , v \in \mathcal{C} \\ 0 & , otherwise \end{cases}
\tag{94}
$$

$$
\psi_{\mathcal{D}}(v) = \begin{cases} \frac{\phi(v)}{vol(\mathcal{D})} & , v \in \mathcal{D} \\ 0 & , otherwise \end{cases}
\tag{95}
$$

$$\psi_{\mathcal{C}\setminus\mathcal{D}}(v) = \begin{cases} \frac{\phi(v)}{vol(\mathcal{C}\setminus\mathcal{D})} & , \ v \in \mathcal{C}\setminus\mathcal{D} \\ 0 & , \ otherwise \end{cases} \tag{96}$$

Therefore,

$$\psi_{\mathcal{C}} = \frac{vol(\mathcal{D})}{vol(\mathcal{C})}\psi_{\mathcal{D}} + \frac{vol(\mathcal{C}\setminus\mathcal{D})}{vol(\mathcal{C})}\psi_{\mathcal{C}\setminus\mathcal{D}} \tag{97}$$

According to the linear property of (lazy) PageRank vectors where $pr(\alpha, s) = \alpha s \sum_{k=0}^{\infty}((1-\alpha)M)^k$, described by definition 16, from equation 97, we further have

$$pr(\alpha, \psi_{\mathcal{C}}) = \frac{vol(\mathcal{D})}{vol(\mathcal{C})}pr(\alpha, \psi_{\mathcal{D}}) + \frac{vol(\mathcal{C}\setminus\mathcal{D})}{vol(\mathcal{C})}pr(\alpha, \psi_{\mathcal{C}\setminus\mathcal{D}})$$
$$\Rightarrow pr(\alpha, \psi_{\mathcal{C}})(\bar{\mathcal{C}}) = \frac{vol(\mathcal{D})}{vol(\mathcal{C})}pr(\alpha, \psi_{\mathcal{D}})(\bar{\mathcal{C}}) + \frac{vol(\mathcal{C}\setminus\mathcal{D})}{vol(\mathcal{C})}pr(\alpha, \psi_{\mathcal{C}\setminus\mathcal{D}})(\bar{\mathcal{C}}) \tag{98}$$

Since $\frac{vol(\mathcal{D})}{vol(\mathcal{C})} + \frac{vol(\mathcal{C}\setminus\mathcal{D})}{vol(\mathcal{C})} = 1$, at least one of the following equations holds.

$$pr(\alpha, \psi_{\mathcal{D}})(\bar{\mathcal{C}}) \leq pr(\alpha, \psi_{\mathcal{C}})(\bar{\mathcal{C}}) \tag{99}$$

$$pr(\alpha, \psi_{\mathcal{C}\setminus\mathcal{D}})(\bar{\mathcal{C}}) \leq pr(\alpha, \psi_{\mathcal{C}})(\bar{\mathcal{C}}) \tag{100}$$

Therefore, for any vertex set $\mathcal{C} \subseteq \mathcal{V}$, and any of its a subset $\mathcal{D} \subseteq \mathcal{C}$. At least one of $pr(\alpha, \psi_{\mathcal{D}})(\bar{\mathcal{C}})$ and $pr(\alpha, \psi_{\mathcal{C}\setminus\mathcal{D}})(\bar{\mathcal{C}})$ is no greater than $pr(\alpha, \psi_{\mathcal{C}})(\bar{\mathcal{C}})$. Hence, if $\mathcal{D}$ is randomly sampled from the subsets of $\mathcal{C}$, with probability at least $\frac{1}{2}$, $pr(\alpha, \psi_{\mathcal{D}})(\bar{\mathcal{C}}) \leq pr(\alpha, \psi_{\mathcal{C}})(\bar{\mathcal{C}})$.

$\square$

**Proof of Theorem 21.**

*Proof.* In Theorem 41, let $\mathcal{C} = \mathcal{S}^*$. Then, since $\mathcal{S} \subseteq \mathcal{S}^*$, from Theorem 42, At least one of $pr(\alpha, \psi_{\mathcal{S}})(\bar{\mathcal{S}}^*)$ and $pr(\alpha, \psi_{\mathcal{S}^*\setminus\mathcal{S}})(\bar{\mathcal{S}}^*)$ is no greater than $pr(\alpha, \psi_{\mathcal{S}^*})(\bar{\mathcal{S}}^*)$. As $\mathcal{S}$ is uniformly randomly sampled from $\mathcal{V}$, the probabilities of sampling $\mathcal{S}$ or $\mathcal{S}^* \setminus \mathcal{S}$ are the same. Since both $\mathcal{S}$ and $\mathcal{S}^* \setminus \mathcal{S}$ has optimal local cluster $\mathcal{S}^*$, with probability at least $\frac{1}{2}$,

$$pr(\alpha, \psi_{\mathcal{S}})(\bar{\mathcal{S}}^*) \leq pr(\alpha, \psi_{\mathcal{S}^*})(\bar{\mathcal{S}}^*) \leq \frac{\Phi(\mathcal{S}^*)}{2\alpha} \Rightarrow pr(\alpha, \psi_{\mathcal{S}})(\mathcal{S}^*) \geq 1 - \frac{\Phi(\mathcal{S}^*)}{2\alpha} \tag{101}$$

$$\therefore pr(\alpha, \psi_{\mathcal{S}})(\mathcal{S}^*) - vol(\mathcal{S}^*) \geq 1 - \frac{\Phi(\mathcal{S}^*)}{2\alpha} - vol(\mathcal{S}^*) \tag{102}$$

According to Theorem 36, $\forall z \in [0, \frac{1}{e}]$, if $\delta = 1 - \frac{\Phi(\mathcal{S}^*)}{2\alpha} - vol(\mathcal{S}^*) > z$, then,

$$\Phi(pr(\alpha, \psi_{\mathcal{S}})) < \sqrt{\frac{9\alpha \ln \frac{1}{z}}{\delta - z}} = \sqrt{\frac{9\alpha \ln \frac{1}{z}}{1 - \frac{\Phi(\mathcal{S}^*)}{2\alpha} - vol(\mathcal{S}^*) - z}} \tag{103}$$

Let $\alpha = \Phi(\mathcal{S}^*) \in [0, 1]$, then,

$$\Phi(pr(\alpha, \psi_{\mathcal{S}})) < \sqrt{\frac{9\alpha \ln \frac{1}{z}}{\delta - z}} = \sqrt{\frac{9\Phi(\mathcal{S}^*) \ln \frac{1}{z}}{\frac{1}{2} - vol(\mathcal{S}^*) - z}} \tag{104}$$

holds for any $z \in [0, \frac{1}{e}]$. When $vol(\mathcal{S}^*) \leq \frac{1}{2}$, we can always find $z \leq \frac{1}{2} - vol(\mathcal{S}^*)$ to bound $\Phi(pr(\alpha, s))$ such that

$$\Phi(pr(\alpha, \psi_\mathcal{S})) < O(\sqrt{\Phi(\mathcal{S}^*)}) \tag{105}$$

Especially, when $vol(\mathcal{S}^*) \leq \frac{1}{3}$, let $z = \frac{1}{18}$,

$$\Phi(pr(\alpha, \psi_\mathcal{S})) < \sqrt{\frac{9\Phi(\mathcal{S}^*)\ln 18}{\frac{1}{2} - vol(\mathcal{S}^*) - \frac{1}{18}}} \leq \sqrt{81\Phi(\mathcal{S}^*)\ln 18} \leq \sqrt{235\Phi(\mathcal{S}^*)} \tag{106}$$

$\square$

## 5 Algorithm Complexity

### 5.1 Worst-case Time Complexity of HyperACL

The pseudo-code of HyperACL is given in Algorithm 2. Assume we have direct access to the support of each $\gamma_e$. Assume the number of hyperedge-vertex connections is $m$, which is the sum of the sizes of the support sets of all $\gamma_e$. Additionally, we assume that the returned local cluster has size $k$.

The computation of $R$ and $W$ takes $O(m)$. The constructed $R$ and $W$ both have $m$ non-zero entries. The construction of $D_\mathcal{V}$ takes $O(|\mathcal{V}|)$ and the construction of $D_\mathcal{E}$ takes $O(|\mathcal{E}|)$. Given that each hyperedge has at least 2 vertices and each vertex has at least one hyperedge incident to it, step 1 takes $O(m)$.

Given that $W$ and $R$ both have $m$ non-zero elements, the multiplication $P = D_\mathcal{V}^{-1} W D_\mathcal{E}^{-1} R$ takes $O(m^2)$ using CSR format sparse matrix. Therefore step 2 takes $O(m^2)$.

Computing the stationary distribution $\phi$ of $P$ takes $O(|\mathcal{V}|^2)$ using power iteration. Therefore, step 1, step 2, and step 3 take $O(m^2)$.

Step 4 takes $O(|\mathcal{V}|^2)$ to calculate the PageRank vector by power iteration. Step 5 takes $O(|\mathcal{V}|log(|\mathcal{V}|))$.

In Step 6, for a local cluster of size $l$, computing its conductance takes $O(l(|\mathcal{V}| - l)$. We record the smallest conductance along computing. Therefore, Steps 6 and 7 take

$$\sum_{l=1}^{k} l \cdot (|\mathcal{V}| - l) = \sum_{l=1}^{k} l|\mathcal{V}| - \sum_{l=1}^{k} l^2 = \frac{k(k+1)}{2}|\mathcal{V}| - \frac{k(k+1)(2k+1)}{6} \in O(k^2|\mathcal{V}|) \tag{107}$$

Note that $m \geq |\mathcal{V}|$. Therefore, the worst-case time complexity of HyperACL is

$$O(m^2) + O(|\mathcal{V}|^2) + O(|\mathcal{V}|log(|\mathcal{V}|)) + O(k^2|\mathcal{V}|) \in O(m^2 + |\mathcal{V}|^2 + k^2|\mathcal{V}|) \tag{108}$$

For GeneralACL, the computation of the transition matrix may not be $P = D_\mathcal{V}^{-1} W D_\mathcal{E}^{-1} R$ but depends on specific data and task properties. Assuming the complexity of computing $P$ is $C$, then the worst-case time complexity of GeneralACL is

$$O(C) + O(|\mathcal{V}|^2) + O(|\mathcal{V}|^2) + O(|\mathcal{V}|log(|\mathcal{V}|)) + O(k^2|\mathcal{V}|) \in O(C + |\mathcal{V}|^2 + k^2|\mathcal{V}|) \tag{109}$$

### 5.2 Worst-case Space Complexity of HyperACL

We make the same assumption as in section 5.1. The storage of $R, W, D_\mathcal{V}, D_\mathcal{E}$ takes $O(m)$. The storage of $P$ takes $O(|\mathcal{V}|^2)$. The storage of stationary distribution and the intermediate results takes $O(|\mathcal{V}|)$. Other steps are not as space-consuming as these.

Therefore, the worst-case space complexity for HyperACL is

$$O(m) + O(|\mathcal{V}|^2) + O(|\mathcal{V}|) \in O(m + |\mathcal{V}|^2) \tag{110}$$

For GeneralACL, Assuming the space complexity of computing $P$ is $C$. The storage of $P$ takes $O(|V|^2)$ and the worst-case space complexity for GeneralACL is

$$O(C) + O(|\mathcal{V}|^2) + O(|\mathcal{V}|) \in O(C + |\mathcal{V}|^2) \tag{111}$$

## 6 Experiments

In this section, we demonstrate the effectiveness of our algorithm on real-world hypergraphs. We first describe the experimental settings and then discuss the experiment results. Details to reproduce the results are provided in Appendix B.

### 6.1 Experimental Setup

**Datasets and Hypergraph Constructions.** We construct 4 citation networks that are subgraphs of AMiner DBLP v14. For hypergraph construction, we use each vertex to represent a scholar and each hyperedge to represent a publication, connecting all its authors. Each hyperedge will be assigned a weight $w(e) = (\# \text{ citations of paper } e) + 1$. Then, realistically, we assign more EDVW for the leading authors and the last authors, and less EDVW for the middle authors. More details on the datasets and hypergraph construction can be found in Appendix B.2.

**Settings and Metrics.** For each dataset, we sample 50 seed vertex sets, which we call observations. For each observation, we first uniformly sample one organization within {MIT, CMU, Stanford, UCB}, then uniformly sample 5 authors (without duplication) affiliated with the sampled organization. Then, we run the methods to get local clusters and compare their quality in terms of (1) low conductance. A good local cluster should be a cohesive set of vertices; (2) a high F1 score with the set of all authors in the sampled organization. In real life, scholars usually coauthor more with others in the same academic institute. Hence, a good cluster should, to some extent, align with the institute label information (Liu et al., 2021).

**Baselines.** We compare our HyperACL with the strong STAR++ and CLIQUE++ expansions, as well as DiffEq. (Takai et al., 2020) which is specifically tailored for non-EDVW hypergraph local clustering. It is proved in (Chitra & Raphael, 2019) that, random walks on non-EDVW graphs are equivalent to some clique graph. More details on baselines are provided in Appendix B.4.

### 6.2 Experimental Results

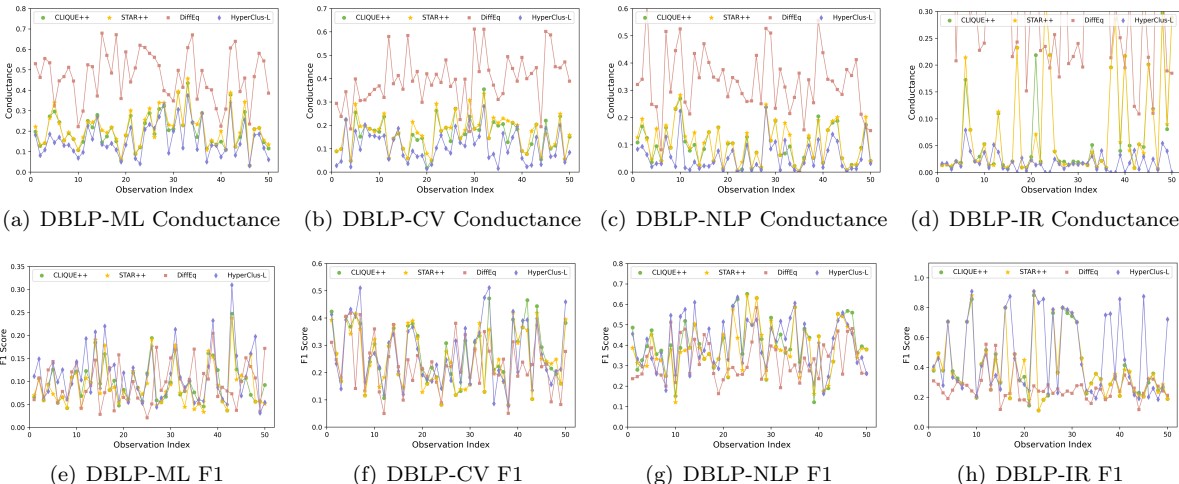

(a) DBLP-ML Conductance    (b) DBLP-CV Conductance    (c) DBLP-NLP Conductance    (d) DBLP-IR Conductance

(e) DBLP-ML F1    (f) DBLP-CV F1    (g) DBLP-NLP F1    (h) DBLP-IR F1

Figure 1: Conductance($\downarrow$) and F1($\uparrow$) Comparison on Local Clustering Task.

Figure 1 shows the local clustering experiment results. **First**, in terms of both conductance and F1 score, HyperACL has the best performance on all the datasets. The purple lines, which represent our HyperACL, are significantly the lowest for conductance and generally the highest for F1. **Second**, CLIQUE++ and STAR++ can achieve sub-optimal results. **Third**, HyperACL is more stable because (1) for almost all the observations, our HyperACL can achieve a smaller conductance; (2) there are several observations in DBLP-IR where our HyperACL achieves a small conductance, but the baseline methods achieve relatively very large conductance.

We report the time per execution for the methods in the local clustering experiments in Figure 2. Generally, CLIQUE++ and HyperACL are better than STAR++ in terms of execution time. Compared to other methods, DiffEq is more stable. DiffEq achieves the best time performance on DBLP-ML, and achieves similar time performance on DBLP-CV and DBLP-NLP. However, DiffEq does not perform well on DBLP-NLP. Generally, our HyperACL has a similar time performance with CLIQUE++.

As a supplement of Figure 1, in Table 2, we report the average conductance and F1 score of the 50 observations on the local clustering task. Our HyperACL achieves the best performance for all the datasets in terms of both metrics.

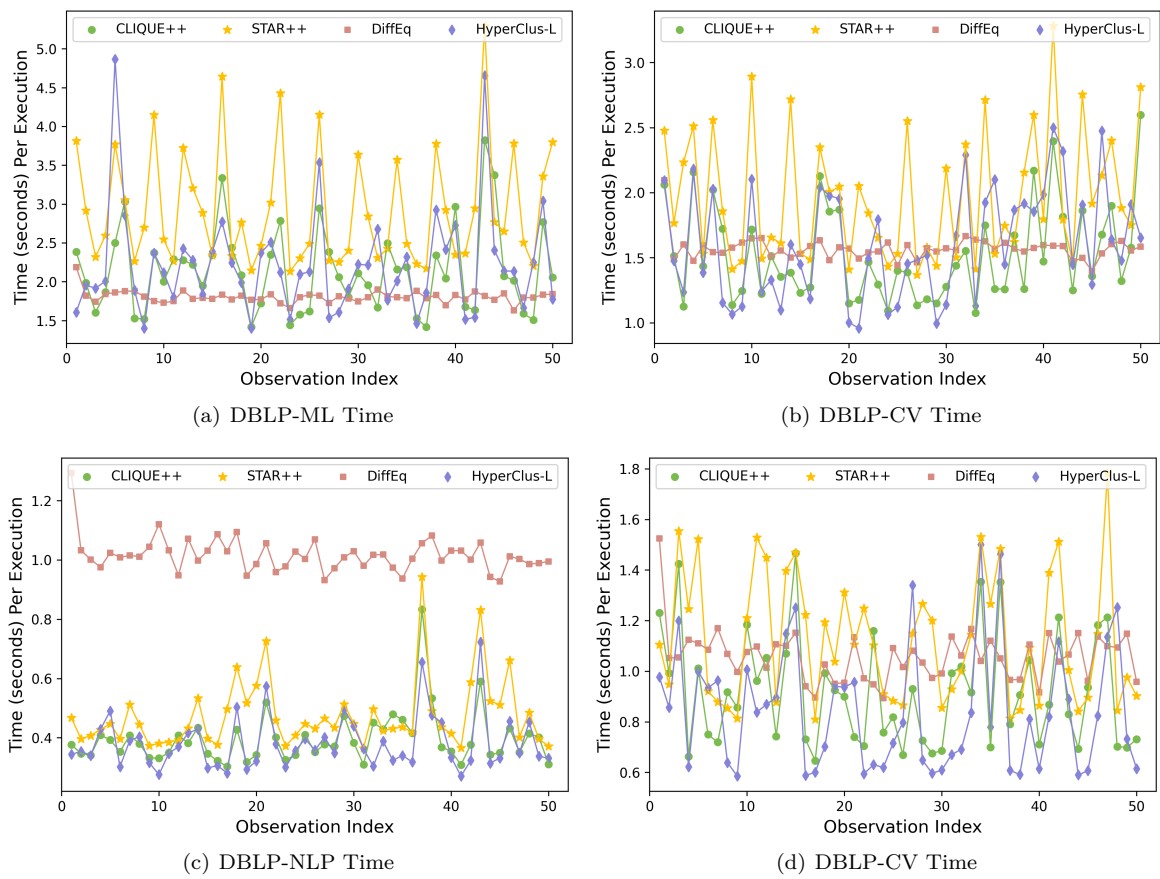

(a) DBLP-ML Time

(b) DBLP-CV Time

(c) DBLP-NLP Time

(d) DBLP-CV Time

Figure 2: Time(↓) Comparison on Local Clustering Task.

# 7 Conclusion

In this work, we extended the Andersen-Chung-Lang (ACL) algorithm to handle weighted, directed, and self-looped graphs, as well as hypergraphs with edge-dependent vertex weights (EDVW). Our proposed algorithms, GeneralACL and HyperACL, demonstrate strong theoretical guarantees, achieving quadratically optimal conductance with at least $\frac{1}{2}$ probability under mild conditions. These contributions address critical

Table 2: Average Conductance(↓) and F1(↑) on Local Clustering.

| Method | DBLP-ML | | DBLP-CV | | DBLP-NLP | | DBLP-IR | |
|---|---|---|---|---|---|---|---|---|
| | $\Phi(\downarrow)$ | F1($\uparrow$) | $\Phi(\downarrow)$ | F1($\uparrow$) | $\Phi(\downarrow)$ | F1($\uparrow$) | $\Phi(\downarrow)$ | F1($\uparrow$) |
| STAR++ | 0.2129 | 0.0918 | 0.1704 | 0.2514 | 0.1045 | 0.3828 | 0.0737 | 0.4038 |
| CLIQUE++ | 0.2012 | 0.0957 | 0.1576 | 0.2620 | 0.0949 | 0.4053 | 0.0766 | 0.3997 |
| DiffEq | 0.4705 | 0.0961 | 0.3966 | 0.2307 | 0.3346 | 0.3259 | 0.3260 | 0.2600 |
| HyperACL | **0.1590** | **0.1125** | **0.1104** | **0.2697** | **0.0559** | **0.4061** | **0.0189** | **0.4802** |

gaps in the literature by generalizing local clustering techniques to more complex graph structures and defining hypergraph conductance from a random walk perspective. Moreover, we introduced an early-stop mechanism to ensure strong locality, significantly improving runtime efficiency in practice. Our experimental results validate the effectiveness of the proposed algorithms, showcasing their applicability across diverse graph and hypergraph datasets.

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

## A  Supplementary Proof

### A.1  Proof of Lemma 17

*Proof.* Recall the definition of $M$ from 16, and given that

$$pr(\alpha, s) = \alpha s + (1 - \alpha)pr(\alpha, S)M \tag{112}$$

it follows that

$$pr(\alpha, s) = \alpha s + \frac{1 - \alpha}{2}pr(\alpha, s) + \frac{1 - \alpha}{2} \cdot pr(\alpha, s) \cdot P \tag{113}$$

and therefore,

$$\frac{1 + \alpha}{2}pr(\alpha, s) = \alpha s + \frac{1 - \alpha}{2}pr(\alpha, s) \cdot P \tag{114}$$

Then, we can give the final expression of $pr(\alpha, s)$ as follows:

$$pr(\alpha, s) = \frac{2\alpha}{1 + \alpha} \cdot s + \frac{1 - \alpha}{1 + \alpha} \cdot pr(\alpha, s) \cdot P \tag{115}$$

On the other hand,

$$rpr\left(\frac{2\alpha}{1 + \alpha}, S\right) = \frac{2\alpha}{1 + \alpha}s + \frac{1 - \alpha}{1 + \alpha}pr(\alpha, s)P \tag{116}$$

which are in the same function of $s, \alpha$ as $pr(\alpha, s)$. So, because of the uniqueness, $pr(\alpha, S)$ is equal to $rpr\left(\frac{2\alpha}{1+\alpha}, S\right)$. It is easy to verify that $\frac{2\alpha}{1+\alpha}$ is a bijective mapping for $\alpha \in [0, 1]$. Therefore, a lazy Random Walk is equivalent to standard PageRank up to a change of $\alpha$.

$\square$

## B  Appendix: Additional Experimental Details

### B.1  Environments

We run all our experiments on a Windows 11 machine with a 13th Gen Intel(R) Core(TM) i9-13900H CPU, 64GB RAM, and an NVIDIA RTX A4500 GPU. One can also run the code on a Linux machine. All the code of our algorithms is written in Python. The Python version in our environment is 3.11.4. In order to run our code, one has to install some other common libraries, including PyTorch, pandas, numpy, scipy, and ucimlrepo. Please refer to our README in the code directory for downloading instructions.

### B.2  Datasets

Table 3 summarize the statistics and attributes of each dataset we used in the experiments.

For local clustering, We first download the newest AMiner DBLP v14 from its official website[3], then filter out the publications with incomplete information, then retrieve all the papers from the year 2018 (inclusive) to the year 2023 (exclusive). Based on this, we retrieve all publications in venues ICML, NIPS, and ICLR to construct DBLP-ML; all publications in venues CVPR, ECCV, and ICCV to construct DBLP-CV; all publications in venues ACL, EMNLP, and NAACL to construct DBLP-NLP; all publications in venues KDD, SIGIR, WWW to construct DBLP-IR. Our datasets are provided within the code of this work.

When filtering out the publications with incomplete information, we remove the publications that have one of the following: (1) any author with an unknown organization. (2) any author with an unknown author ID. The author ID, which is a unique string for each author, is different from the vertex ID. Given the fact that lots of publications have missing information, the final datasets that we are using contain only a portion of actual publications in the venues.

---

[3]https://www.aminer.org/citation

According to the statistics from Paper Copilot, from the year 2018 to the year 2023, 4895 papers were accepted by ICML[4], 3481 papers were accepted by ICLR[5], 9344 papers were accepted by NeurIPS[6]. The total number of vertices in the DBLP-ML datasets should be 17720, and we have 6617 ($\approx$ 37.3%) in our dataset.

According to the public GitHub repository about conference acceptance[7], from the year 2018 to the year 2023, 7471 papers were accepted by CVPR, 2694 papers were accepted by ICCV, and 3782 papers were accepted by ECCV. The total number of vertices in the DBLP-CV dataset should be 13947, and we have 4287 ($\approx$ 31%).

According to the same GitHub repository about conference acceptance, from the year 2018 to the year 2023, 3151 papers were accepted by ACL, 3868 papers were accepted by EMNLP, 1395 papers were accepted by NAACL. The total number of vertices in the DBLP-NLP dataset should be 5991, and we have 1728 ($\approx$ 29%) in our data set.

According to the same GitHub repository about conference acceptance, from the year 2018 to the year 2023, 1215 papers were accepted by KDD, 629 papers were accepted by SIGIR, and 1293 papers were accepted by WWW. The total number of vertices in the DBLP-IR dataset should be 3137, and we have 2848 ($\approx$ 91%). There is no author in UCB that coauthors any paper in our DBLP-IR (actually, according to CS Rankings[8], only 4 authors and 5 papers were accepted into KDD, SIGIR or WWW from the year 2018 to the year 2023). Therefore for our DBLP-IR, instead of first uniformly sampling one seed institute from {MIT, CMU, Stanford, UCB}, we uniformly sample one from {MIT, CMU, Stanford}.

Table 3 shows the statistics of our datasets used in the local clustering experiments. The meanings of columns are the name of the hypergraphs, number of vertices/authors, number of hyperedges/publications, number of authorship connections, number of edges in the clique expansion graphs, number of organizations, number of authors affiliated with Massachusetts Institute of Technology, number of authors affiliated with Carnegie Mellon University, number of authors affiliated with Stanford University, number of authors affiliated with UC Berkeley, and the included venues.

**Edge-Dependent Vertex Weight Assignment on Citation Hypergraphs for Local Clustering.**
For a publication in computer science, usually, the middle authors have fewer contributions to the paper than the authors in the front (who are usually the principal investigators) and the authors in the back (who are usually advisors who manage the project). Therefore, for each publication/hyperedge, we exponentially assign more vertex weights for the authors/vertex in the front or in the back. Specifically,

(1) For a publication $e$ that has an odd number $(2k + 1)$ of authors indexed by $0, 1, ..., 2k$, we assign vertex weight 1 for the middle author (indexed by $k$), and multiply the vertex weight by 2 when sweeping to the front or the back:

$$\gamma_e(x) = 2^{|x-k|}, \forall x \in \{0, 1, ..., 2k\} \tag{117}$$

(2) For a publication $e$ that has an even number $(2k)$ of authors indexed by $0, 1, ..., 2k-1$, we assign vertex weight 1 for the middle two authors (indexed by $k-1, k$), and multiply the vertex weight by 2 when sweeping to the front or the back:

$$\gamma_e(x) = \begin{cases} 2^{k-1-x}, & \forall x \in \{0, 1, ..., k-1\} \\ 2^{x-k}, & \forall x \in \{k, k+1, ..., 2k\} \end{cases} \tag{118}$$

---

[4]https://papercopilot.com/statistics/icml-statistics/
[5]https://papercopilot.com/statistics/iclr-statistics/
[6]https://papercopilot.com/statistics/neurips-statistics/
[7]https://github.com/lixin4ever/Conference-Acceptance-Rate
[8]https://csrankings.org/#/fromyear/2018/toyear/2022/index?inforet&us

Table 3: Statistics of Constructed Hypergraphs in Local Clustering Experiments

| Hypergraph | $|\mathcal{V}|$ | $|\mathcal{E}|$ | $\sum_{v \in \mathcal{V}} E(v)$ | $|\mathcal{E}|$ clique | # orgs | MIT | CMU | Stanford | UCB | included venues |
|---|---|---|---|---|---|---|---|---|---|---|
| DBLP-ML | 14958 | 6617 | 25790 | 83462 | 5117 | 462 | 409 | 478 | 520 | ICML, NeurIPS, ICLR |
| DBLP-CV | 12116 | 4287 | 19883 | 76360 | 3813 | 115 | 175 | 128 | 135 | CVPR, ECCV, ICCV |
| DBLP-NLP | 4555 | 1728 | 6907 | 24346 | 1165 | 42 | 146 | 45 | 32 | ACL, EMNLP, NAACL |
| DBLP-IR | 8601 | 2848 | 12203 | 48965 | 3409 | 18 | 72 | 66 | 0 | KDD, SIGIR, WWW |

## B.3 Metrics

The F1 score between two sets $\mathcal{A}_m$ and $\mathcal{A}_c$ is defined as

$$
\begin{aligned}
&TP\textit{(True Positive)} = |\mathcal{A}_m \cap \mathcal{A}_c| \\
&FP\textit{(False Positive)} = |\mathcal{A}_m \setminus \mathcal{A}_c| \\
&FN\textit{(False Negative)} = |\mathcal{A}_c \setminus \mathcal{A}_m| \\
&precision = \frac{TP}{TP + FP} \\
&recall = \frac{TP}{TP + FN} \\
&F1 = \frac{2 \times precision \times recall}{precision + recall}
\end{aligned}
\tag{119}
$$

Assume for an observation, all the authors in the academic institute we sampled construct vertex set $\mathcal{A}_c$, and the algorithm returns a local cluster $\mathcal{A}_m$. We compute the conductance of vertex set $A_m$ as Definition 14.

$$
\Phi(\mathcal{A}_m) = \frac{|\partial \mathcal{A}_m|}{\min(vol(\mathcal{A}_m), vol(\bar{\mathcal{A}_m}))} = \frac{|\partial \mathcal{A}_m|}{\min(vol(\mathcal{A}_m), 1 - vol(\mathcal{A}_m))}
\tag{120}
$$

Then, we compute the F1 score between $A_m$ and $A_c$ as Equation 119.

## B.4 Baselines

**CLIQUE++.** For each hyperedge $e$, each $u, v \in e$ with $u \neq v$, we add an edge $uv$ of weight $w(e)$. Then, we apply the ACL algorithm Andersen et al. (2006) to compute the sweep cuts of the weighted clique graph. We use exact PageRank vector instead of an approximated one, and adopt the same early-stop mechanism described at the end of section 3.

**STAR++.** For each hyperedge $e$, we introduce a new vertex $v_e$. For each vertex $u \in e$, we add an edge $uv_e$ of weight $w(e)/|e|$. After converting the hypergraph into a star graph, we do the same algorithm for local clustering as in CLIQUE++.

**DiffEq** Takai et al. (2020). We directly use the official code[9] of this algorithm. This method sweeps over the sweep sets obtained by differential equations for local clustering. For global partitioning, it simply calls local clustering for every vertex and returns the best in terms of conductance. Originally, this algorithm could only take one starting vertex for local clustering. We modified the code to add one additional vertex that connects the 5 starting vertices in each observation. Then regard the newly added vertex as the starting vertex, so that it can obtain the local cluster for the given 5 starting vertices.

In local clustering experiments, since in Theorem 21, $\alpha = \Phi(\mathcal{S}^*)$ is agnostic because the obtaining the optimal $\mathcal{S}^*$ is NP. We run our HyperACL two times. For the first time, we take $\alpha = \Phi(\mathcal{S})$, the conductance of the starting vertex set, and obtain a local cluster $\mathcal{S}'$. Then, we take $\alpha = \Phi(\mathcal{S}')$ and run the algorithm again to

---

[9]https://github.com/atsushi-miyauchi/Hypergraph_clustering_based_on_PageRank

obtain a local cluster as the result. For CLIQUE++ and STAR++, we directly pass $\Phi(\mathcal{S}')$ obtained from HyperACL to them as the $\alpha$ value. For a fair comparison, we do not time the first execution of HyperACL, which obtains a good $\alpha$ for all algorithms to use.

