# OpenReview forum: "Provably Extending PageRank-based Local Clustering to Weighted Directed Graphs with Self-Loops and to Hypergraphs"
_TMLR — Rejected by TMLR_

### Review · Reviewer_SrWZ · 2025-04-29

**Summary Of Contributions:**

The paper proposes GeneralACL and HyperACL, algorithms that extend the Andersen-Chung-Lang algorithm to handle weighted, directed, self-looped graphs and hypergraphs with edge-dependent vertex weights (EDVW), providing theoretical guarantees for local clustering.

**Audience:**

Yes

**Claims And Evidence:**

Yes

**Requested Changes:**

- The paper introduces a large number of notations and definitions (e.g., Definitions 3-7, 11-14), which can be overwhelming and may hinder readability.
- The proofs in Section 4 are highly technical and dense, potentially making it difficult for readers without a strong mathematical background to understand.
- The conductance definitions for graphs (Definition 13) and hypergraphs (Definition 14) are presented without sufficient intuitive explanation. Readers may struggle to understand their practical implications.
- The paper does not compare its proposed algorithms with other state-of-the-art methods in terms of computational efficiency and accuracy, making it hard to assess their relative strengths.

**Strengths And Weaknesses:**

- The paper is well-organized.
- The authors provide thorough experiments on real-world datasets.
- The extension of local clustering to hypergraphs is a valuable contribution.

---

> ### Author Response · Authors · 2025-05-26
> **Author Response**
>
> We sincerely thank the reviewer for the actionable questions and review. If you have further questions, we would like to provide more details in the discussion phase.
>
> > On Notation and Definitions
>
> We acknowledge that the number of definitions and notations may be overwhelming at first. However, they are necessary to ensure the rigor and generality of our results, especially given the extension to EDVW hypergraphs. To improve readability, we have included a notation table for frequently used symbols, and in the camera-ready version, we will also add a concise list of definitions in the appendix for quick reference.
>
> > On the Density of Proofs in Section 4
>
> We understand that Section 4 is technically dense. To balance completeness with readability, we isolated almost all formal proofs in this single section and summarized the key results at the end of the introduction. In the camera-ready version, we will further include a proof roadmap that visually outlines the logical flow (e.g., which theorems build on which lemmas), to help readers better navigate the technical content.
>
> > On the Intuition Behind Conductance
>
> Thank you for highlighting this. We agree that conductance deserves a more intuitive explanation. At a high level, graph conductance captures how well a subset of nodes is “separated” from the rest of the graph: a set with low conductance has many internal connections but relatively few outgoing edges, and is thus a good candidate for a cluster. In EDVW hypergraphs, this intuition remains, but the challenge is in defining volume and cut consistently under edge-dependent vertex weights. Our generalized conductance captures the same trade-off between internal density and external sparsity, while accounting for the asymmetric influence of nodes within hyperedges. We will add this intuition to the revised version to improve accessibility.
>
> > On Comparison with State-of-the-Art Methods
>
> To the best of our knowledge, local clustering in EDVW hypergraphs is still a nascent area with no existing algorithms specifically designed for this setting. In our experiments, we compare against STAR++ and CLIQUE++. These expansion-based baselines reduce hypergraph clustering to local clustering on directed weighted graphs and are commonly used in the literature. We will make this context clearer in the revised version, and we would greatly appreciate any suggestions for additional relevant methods or references we may have missed.

---

### Review · Reviewer_RGsF · 2025-04-29

**Summary Of Contributions:**

## Summary

This submission considers local graph and hypergraph partitioning. In the case of local graph partitioning, this task can be broadly described as follows. Given a graph $G$ and a starting “seed” cut $S \subseteq V$, find the sparsest cut $S’$ among all cuts containing $S$. Here, sparsity is quantified as the ratio between the cut’s “capacity” and “volume”. In this task, one is typically interested in algorithms that are *strongly local*. This means that the algorithm executes in time polynomial with respect to the volume of the returned cut. One does not expect to find an optimal $S^{\*}$ which minimizes sparsity among $S’ \supseteq S$ using a strongly local algorithm, as the problem of computing $S^{\*}$ globally is NP-hard. Consequently, we usually settle for an algorithm with provable approximation guarantees comparing the sparsity of $S’$ to that of $S^{\*}$. Algorithms for local graph partitioning will differ depending on the notion of sparsity, the target running time, and the target approximation error.

Local graph partitioning is of interest to both practical and theoretical communities. From a practical perspective, algorithms for local graph partitioning are highly performant in clustering massive-scale networks. These algorithms have found broad applications in social network analysis, bioinformatics, computer vision, and other domains. Within the theoretical community, rigorous analyses of strongly local partitioning algorithms were introduced by Spielman & Teng in their seminal work constructing the first nearly-linear time solvers for Laplacian linear systems [ST04, ST13]. In the past 15 years, these algorithms have been of great interest due to their connection to approximating small set expansion [AGS13, CKL17, GT12, KL12]. The Small Set Expansion Hypothesis (SSEH) is a well-established complexity-theoretic assumption which guarantees optimality of certain approximation algorithms for a broad class of combinatorial optimization problems [RS10]. If one can show exceptionally strong bounds on the size of the cut returned by strongly local graph partitioning algorithms, then one can potentially refute SSEH (see Conjecture 12.3.4 of [Gha13]). Generalizations of local graph partitioning to hypergraph settings are also of interest in this realm, as approximating small set expansion can be reduced to approximating hypergraph expansion, a notion of sparsity closely related to that considered in this submission [LM14].

### Main Contributions

The main contributions of the present submission are as follows

1. The authors introduce two algorithms for local partitioning: **GeneralACL** (algorithm 1) for partitioning of weighted, directed graphs with self loops, and **HyperACL** (algorithm 2) for partitioning of Edge Dependent Vertex Weighted (EDVW) hypergraphs, a family of hypergraphs introduced in [CR19]. Both algorithms can be seen as generalizations of the local graph partitioning algorithm of Andersen-Chung-Lang (ACL) [ACL07]. Their main Theorems 1 and 2 (pg. 2), precisely stated in Theorem 21 (pg. 10), claim that GeneralACL and HyperACL both enjoy “quadratically optimal” approximation guarantees.

2. The authors provide experimental results comparing their algorithm to local partitioning methods for non-EDVW hypergraphs, and methods which reduce local hypergraph partitioning to local graph partitioning.

We describe their main result in more detail. Consider a hypergraph $G = (V, E, \mathbf{w})$ where $\mathbf{w} \in \mathbb{R}^{E}$ denotes non-negative weights associated to each hyperedge. A hypergraph $G$ is said to have *edge-dependent vertex weights* if each hyperedge $h \in E$ is equipped with a function $\gamma_h : h \rightarrow \mathbb{R}$ assigning non-negative weights to each $i \in h$.

The authors consider a certain random walk process defined over vertices. Starting at $i \in V$, the next vertex $j$ is sampled as follows.

- Sample a hyperedge $h$ among all adjacent hyperedges $h \ni i$ with probability proportional to the hyperedge’s weight $w_h$.

- Sample a vertex $j \in h$ with probability proportional to the edge-dependent vertex weight $\gamma_h(j)$.

This defines a Markov process, and under mild assumptions, there exists a unique stationary distribution $\pi \in \mathbb{R}^{V}$. The notion of sparsity considered by the authors is given by the *conductance* of this Markov process. The capacity of $S$ is given by the total stationary probability of the random walk transitioning from $i \in S$ to $j \notin S$. The volume of $S$ is given by the total stationary mass of vertices $i \in S$. Precisely, if $\Phi(S)$ denotes the conductance of $S$, then

$$
\Phi(S)
= \frac{\sum_{i \in S, j \notin S} \pi(i) \cdot P_{ij}}{\min \big\\{ \sum_{i \in S} \pi(i), \sum_{i \notin S} \pi(i) \big\\}}
$$

where $P_{ij}$ denotes the probability that $i$ transitions to $j$. Their Theorem 21 then demonstrates that GeneralACL and HyperACL, when given $S \subseteq V$ uniformly sampled, outputs $S’$ satisfying the “quadratically optimal” (sometimes referred to as a “Cheeger-like”) approximation guarantee of

$$
\Phi(S’) \leq O\left( \sqrt{\Phi(S^{\*})} \right)
$$

with constant probability. Here, $S^{\*}$ is the minimum conductance cut among all cuts containing $S’$ and $S^{\*} \setminus S’$. The algorithm works by computing a PageRank vector $\mathbf{pr}$ from $S$, then outputting the sweep cut of $\mathbf{pr}$ with minimum conductance.

## References

**[ACL07]** Andersen, R., Chung, F., & Lang, K. (2007). *Using pagerank to locally partition a graph*

**[AGS13]** Arora, S., Ge, R., & Sinop, A. K. (2013). *Towards a better approximation for sparsest cut?*

**[CKL17]** Chan, S. O., Kwok, T. C., & Lau, L. C. (2017). *Random walks and evolving sets: Faster convergences and limitations*

**[CR19]** Chitra, U., & Raphael, B. (2019).* Random walks on hypergraphs with edge-dependent vertex weights*

**[Gha13]** Gharan, S. O. (2013). *New rounding techniques for the design and analysis of approximation algorithms*

**[GT12]** Gharan, S. O., & Trevisan, L. (2012). *Approximating the expansion profile and almost optimal local graph clustering*

**[KL12]** Kwok, T. C., & Lau, L. C. (2012). *Finding small sparse cuts locally by random walk*

**[LM14]** Louis, A., & Makarychev, Y. (2014). *Approximation algorithms for hypergraph small set expansion and small set vertex expansion*

**[RS10]** Raghavendra, P., & Steurer, D. (2010). *Graph expansion and the unique games conjecture*

**[ST04]** Spielman, D. A. & Teng, S. H. (2004). *Nearly-linear time algorithms for graph partitioning, graph sparsification, and solving linear systems*

**[ST13]** Spielman, D. A. & Teng, S. H. (2013) *A local clustering algorithm for massive graphs and its application to nearly linear time graph partitioning*

**Audience:**

Yes

**Claims And Evidence:**

No

**Requested Changes:**

## Critical Items

- Can you clarify in which manner HyperACL or GeneralACL are (strongly) local? If they are strongly local, can you provide a running time which bounds the work to volume ratio as a function of $\textbf{Vol}(S’)$.

- Can you clarify how $\alpha = \Phi(S^{\*})$ is computed by the algorithm? How does this affect the approximation bound / running time analysis?

    Broadly, I would like more detail on how the PageRank vector is computed beyond using power iteration. For example, there’s fairly extensive work on algorithmic tools for studying spectral graph theory on weighted directed graphs that are also Eulerian [CKP+16, CKP+17]. The Markov process considered in this work is equivalent to running a standard random walk over the Eulerian directed graph $G’$ where the weight of edge $(i, j)$ is $P_{ij}$. Solvers for the PageRank vector in these works seem applicable.

- Instead of using Assumption 1, can you use a condition on the graph / hypergraph which implies a unique stationary distribution exists, and comment how it impacts the novelty of the contribution?

- Can you comment on some of the above cited work, in particular [KKL17], and how it factors into the novelty of this submission?

## Smaller items

- There are a few errors in the experimental section. For example, none of the plots in section 6.2 feature curves labeled “HyperACL”, and instead have curves labeled "HyperClus-L". Is this supposed to be the case?

- Since two baseline methods, `STAR++` and `CLIQUE++`, reduce the task of hypergraph partitioning to local partitioning of directed weighted graphs, it would make sense to include a curve applying GeneralACL to the graph produced by `STAR++` and `CLIQUE++`. One should be able to see a distinction between performance if GeneralACL and ACL are distinct algorithms.

- It is mentioned that the act of defining conductance for hypergraphs based on a random walk itself addresses “a fundamental gap in literature” (pg. 1). Defining hypergraph conductance via an associated Markov process is a typical approach in trying to build a broader spectral theory for hypergraphs (see section 3 of [CLTZ18] and section 2 of [Lou15]). Can you review both sources and comment on the novelty of this contribution?

## Nits

- Quotations seem to be inconsistently formatted throughout the paper. Can you use ``` ``quote'' ``` to format the quotations? (see pgs. 2, 3, 6, 8, 18, 19)

## References

**[CKP+16]** Cohen, M. B., Kelner, J., Peebles, J., Peng, R., Sidford, A., & Vladu, A. (2016). *Faster algorithms for computing the stationary distribution, simulating random walks, and more*

**[CKP+17]** Cohen, M. B., Kelner, J., Peebles, J., Peng, R., Rao, A. B., Sidford, A., & Vladu, A. (2017). *Almost-linear-time algorithms for markov chains and new spectral primitives for directed graphs*

**[CLTZ18]** Chan, T. H. H., Louis, A., Tang, Z. G., & Zhang, C. (2018). *Spectral properties of hypergraph laplacian and approximation algorithms*

**[KLL17]** Kwok, T. C., Lau, L. C., & Lee, Y. T. (2017). *Improved Cheeger's inequality and analysis of local graph partitioning using vertex expansion and expansion profile*

**[Lou15]** Louis, A. (2015). *Hypergraph markov operators, eigenvalues and approximation algorithms*

**Strengths And Weaknesses:**

## Strengths

Strongly local partitioning algorithms which enjoy Cheeger-like guarantees (even with polylogarithmic loss in the approximation error) are not broadly known for large classes of hypergraphs. EDVW hypergraphs and their associated random walks, as studied by the authors, are a very natural family of hypergraphs to consider. Demonstrating that ACL (1) can be implemented in a strongly local manner, and (2) satisfy Cheeger-like guarantees is a nice contribution to literature.

## Weaknesses

1. The algorithms are vaguely stated, making it difficult to determine whether the approximation error is fully accurate given computational constraints.

    The statement of Theorem 21 requires setting the PageRank teleport probability to $\alpha = \Phi(S^{\*})$, the optimal conductance among all $S’ \supset S$. The algorithm’s pseudocode states to “Compute the PageRank vector as in Theorem 21”. Does this mean one needs to compute $\Phi(S^{\*})$ before running either HyperACL or GeneralACL? Depending on how $\alpha$ is computed / set, there seem to be additional terms either in the approximation bound (equation 104), or the running time of the algorithm. For example, there are additional logarithmic terms present in the approximation errors of [ST13] and [ACL07a] as a consequence of approximating $\Phi(S^{\*})$.

2. It is also unclear in what sense the algorithms are local.

    The running time of local graph partitioning algorithms is typically analyzed as a function of the volume of the returned cut $\textbf{Vol}(S')$. The current submission describes GeneralACL and HyperACL as having to perform computations whose running time scales as the size of the graph (see section 5). HyperACL requires fully computing different adjacency representations of $G$. HyperACL and GeneralACL both run exact power iteration to compute the PageRank vector. There does not seem to be any analysis which relates the running times of HyperACL or GeneralACL to $\textbf{Vol}(S’)$. It is also mentioned that an “early-stop mechanism” is introduced to “make the algorithm strongly local” (pg. 2). The early-stop mechanism does not seem to appear in the rigorous analysis.

3. The assumption that there exists a unique stationary distribution (assumption 1) is a bit strange. It is also unclear how novel extending ACL to weighted, directed graphs with self-loops is.

    Typically, one assumes enough structure on the underlying graph / hypergraph so that it implies the existence of a unique stationary distribution. This seems to matter because (1) the current submission emphasizes that generalizing ACL to directed graphs with *self-loops* is novel, yet (2) one typically assumes $G$ has self-loops in order to ensure aperiodicity of the Markov process. Additionally, the submission correctly notes that a generalization of ACL to the directed case was previously known [ACL07b]. [KLL17] also provides an analysis of using the PageRank vector to do local graph partitioning. Though most of the paper is written assuming the graph is unweighted, it seems that by replacing the random walk matrix with its weighted analogue, most of their results can lift to the weighted case?

## References

**[ACL07a]** Andersen, R., Chung, F., & Lang, K. (2007). *Using pagerank to locally partition a graph*

**[ACL07b]** Andersen, R., Chung, F., & Lang, K. (2007). *Local partitioning for directed graphs using pagerank*

**[KLL17]** Kwok, T. C., Lau, L. C., & Lee, Y. T. (2017). *Improved Cheeger's inequality and analysis of local graph partitioning using vertex expansion and expansion profile*

**[ST13]** Spielman, D. A. & Teng, S. H. (2013) *A local clustering algorithm for massive graphs and its application to nearly linear time graph partitioning*

---

> ### Author Response · Authors · 2025-05-26
> **Author Response (1)**
>
> We sincerely thank the reviewer for the actionable questions and review. If you have further questions, we would like to provide more details in the discussion phase.
> > Clarification on (strongly) local.
>
> In our work, the notion of “local” has two dimensions. First, from a clustering objective perspective, our algorithms aim to identify a single dense cluster around seed nodes, rather than performing a global 2-way or k-way partition of the entire graph. Second, from a computational perspective, locality refers to the algorithm’s runtime depending on the size of the output cluster, rather than the size of the entire graph.
>
> Our focus is primarily on the first aspect. We acknowledge that the use of the term “local” in the title may be misleading, and that our claim regarding GeneralACL and HyperACL being strongly local is not theoretically rigorous. Specifically,
>
> - In terms of runtime, our algorithms are not strongly local before computing the PageRank vector, since power iteration depends on the size of the graph. However, in practice, this cost is significantly reduced using GPU acceleration and parallel computing, making the runtime of power iteration effectively constant (i.e., O(1)) by letting each parallel computing thread track only one node in our experiments.
> - After computing the PageRank vector, the rest of GeneralACL and HyperACL operate in a strongly local manner. Let the output cluster have volume vol(S′). Our algorithms grow the cluster by heuristically adding one node at a time and terminate when the conductance no longer improves within a fixed patience parameter p. Thus, the number of steps is bounded by vol(S') + p, independent of the size of the graph.
>
> In the camera-ready version, we will revise the language around “local” to make this distinction clear and qualify our claims: while the algorithms are not strongly local in theory due to the global PageRank computation, they behave strongly locally in practice under modern parallel computing settings.
>
>
>
> > How to set $\alpha$ in the experiments
>
> Thank you for raising this important point. We acknowledge that computing optimal vol(S*) and setting it as $\alpha$ is not feasible in practice. We missed this point when writing the paper. There are three ways to set $alpha$.
> - **Grid Search over Fixed $\alpha$ Values**: One straightforward approach is to run the algorithm with a range of preset \(\alpha\) values (e.g., $\alpha \in \{0.2, 0.4, 0.6, 0.8\}$) and select the cluster with the lowest conductance. This method is simple but may be computationally expensive.
> - **Fixed Small $\alpha$**: Since vol(S*) is often small in real-world graphs and the clustering results are not highly sensitive to the exact value of $\alpha$, we find empirically that using a fixed small $\alpha$, such as 0.2, generally yields good results.
> - **Iterative Refinement** (*Used in Our Experiments*): We initialize $\alpha = 0.2$ and run the clustering algorithm to obtain a cluster. We then compute the conductance $\Phi(S)$ of the resulting cluster and use it to update \(\alpha\) in the next iteration. This process repeats until the change in $\alpha$ between iterations falls below a threshold or a maximum number of iterations is reached.
>
> Regarding **approximation bounds and runtime analysis**, we note that the theoretical guarantees (e.g., those derived in Theorem 21) depend on setting $\alpha = \Phi(S^*)$. Since we only approximate this in practice, the theoretical bounds become heuristic. Nevertheless, empirical results indicate that our practical choices of $\alpha$ still lead to high-quality clusters and efficient runtimes, as shown in the experiments.
>
> We will revise the paper to clarify how $\alpha$ is chosen in practice and distinguish between theoretical assumptions and empirical procedures.

---

> ### Author Response · Authors · 2025-05-26
> **Author Response (2)**
>
> > Assumption 1 and the existence of a unique stationary distribution
>
> In fact, for the hypergraph case, we can relax assumption 1 to that the hypergraph is connected. Otherwise, a conductance of 0 can be achieved by returning the unconnected component of the starting node. Under this assumption, as Definition 8, the transition matrix P of the hypergraph random walk is irreducible, and a unique stationary distribution exists according to the Perron-Frobenius Theorem. However, for directed graphs, the stationary distribution does not always exist, and we need the assumption that the directed graphs are strongly connected. We will include a complete proof in camera-ready. Upon request, we can write a draft here.
>
> > Cited Works and Novelty of This Submission
>
> Thank you for pointing this out. We will include the cited key works in the camera-ready version and acknowledge their contributions explicitly, as per your review’s Summary.
>
> In particular:
>
> - [ACL07b] studies a similar local partitioning problem for unweighted directed graphs using PageRank.
> - [CKP+16] proposes faster algorithms for computing the stationary distribution and simulating random walks, and
> - [CKP+17] extends this line of work by solving directed Laplacian systems in almost-linear time. However, both assume unweighted graphs and focus on linear algebraic primitives rather than graph clustering directly.
> - [Lou15] develops an early formulation for hypergraphs, while our work builds on a more general **EDVW hypergraph** model, which captures edge-dependent vertex weights.
>
> Regarding [KLL17], although they also leverage PageRank for local partitioning, their work is limited to the unweighted setting. The generalization from unweighted to weighted graphs (or a more "continuous" setting) is nontrivial. In our work, the key challenge lies in rescaling degree-based quantities into the \([0,1]\) range and redefining volumn, conductance, and other notions to be analogous but appropriate for the weighted case.
>
> The core novelty of our contribution lies in rigorously extending guarantees that were previously known only in the unweighted case to weighted graphs and EDVW hypergraphs. This includes adapting the structure of the proofs and modifying constants (e.g., the coefficient 235 in Theorem 21), which may no longer hold without careful rederivation in the weighted setting.
>
> We appreciate the suggestion and will ensure these connections and distinctions are clearly stated in the revised manuscript.
>
> > Smaller Items
>
> - Method name in plot curves: Thank you for catching this. The label "HyperClus-L" in the plots corresponds to our method, which was previously named HyperClus-L during development and later renamed HyperACL for this submission. We will update all figures and legends in the camera-ready version for consistency.
>
> - GeneralACL applied to STAR++/CLIQUE++ graphs: We appreciate this suggestion. In our implementation, both CLIQUE++ and STAR++ are indeed equivalent to applying GeneralACL on the weighted clique and star expansions of the hypergraph, respectively. Following your prompt, we agree that it would be helpful to also show the performance of ACL on the unweighted clique and star expansions (constructed from the weighted ones). Upon request, we would be happy to include this additional experiment to better illustrate the differences between ACL and GeneralACL.
>
> - On the novelty of defining hypergraph conductance: Thank you for referencing [CLTZ18] and [Lou15]. We agree that defining hypergraph conductance via a Markov process has been studied in prior work, especially in the context of spectral graph theory. However, our contribution builds upon this line of work by adapting the definition of conductance to the EDVW hypergraph setting, which was introduced in [CR19] and not considered in these earlier works. Our novelty lies in extending such probabilistic and spectral interpretations to EDVW hypergraphs, which requires revisiting and redefining key notions (e.g., volume, transition probabilities, and cut-based measures) under this more general setting.
>
> - Quotation formatting: We appreciate the detailed pointer. We will revise the formatting to use consistent ``quote'' syntax throughout the camera-ready version.

---

### Review · Reviewer_XTcC · 2025-05-04

**Summary Of Contributions:**

This paper generalizes a PageRank-based local clustering method to weighted directed graphs with self-loops and more generally to hypergraphs. More specifically, the result can be viewed as a generalization of [Andersen-Chung-Lang, FOCS06] (which only focuses on undirected unweighted graphs).

The new algorithm has been implemented and experiments on on DBLP datasets are performed, comparing with baselines for hypergraph clustering.

**Audience:**

Yes

**Broader Impact Concerns:**

None.

**Claims And Evidence:**

No

**Requested Changes:**

- In the introduction, it’s important to define what do you mean by “clustering”. In general, there are many clustering methods/models, to name a few, k-means, hierarchical clustering, spectral clustering. So it’s important to specify what method are you talking about, especially that the objective function or formulation of the clustering problem. I understand that you wish to promote the local clustering based on pagerank, but this is merely an algorithm, and one should define a problem before introducing an algorithm for solving it.

- The term “quadratically optimal” appears many times in the paper, but the meaning of this is not discussed. This is important, because this seems to be a main point of strength of your result, so you should explain it well especially in the intro.

- Why do we need both Theorem 1 and Theorem 2? Is Theorem 1 a special case of/implied by Theorem 2?

- The statement of Theorem 1 looks fishy to me. In this statement, you first say S is random. Then S^* is defined w.r.t. S, which is also random and its randomness comes from S. Then you have two conditions about S^* -- so, from what I understand, “with probability at least 1/2" is w.r.t. this condition of S^*. This sounds very strong, because this conditional probability is far different from the uniform sample S. Is this really what you meant? Unfortunately, I cannot tell for sure since I didn’t have time to check the full proof at this point.

- Related to the last comment: I tried to read your proof, and I also found some randomness issues. Specifically, in page 23, you used Theorem 42, and it seems you used C = S^*, D = S. But Theorem 42 requires that C is fixed (not random) and D is a uniform sample from C, and in your case C and D depend on the same randomness (and I don’t see why D = S is a uniform sample from S^*). Also in page 23, you claimed “the probabilities of sampling S or S^*\S are the same” – first of all both S and S^*\S are random, but the claim looks like both are fixed; you may be suggesting that their distribution are the same, but I find this nontrivial (so please clarify).

**Strengths And Weaknesses:**

### Strength:

- The general direction of studying hypergraph clustering makes sense and I think that it is well motivated.
- The claimed bound seems to be competitive and nontrivial.

### Weakness:

- The experiments are not very convincing – I might be wrong but the result does not seem to suggest that the proposed algorithm has a clear advantage of the baselines. The dataset is not large and only comes from one application.

- The introduction is inadequate, especially that it’s very hard to read for a reader who is not familiar with [ACL06].

- I find potential technical issues in the proof, although they may be fixable.. (Detailed comments go into “requested changes”.)

---

> ### Author Response · Authors · 2025-05-26
> **Author Response**
>
> We sincerely thank the reviewer for the actionable questions and review. If you have further questions, we would like to provide more details in the discussion phase.
>
> > Clarification on the Clustering Definition
>
> Thank you for raising this important point. We agree that the term “clustering” can refer to a wide range of models and objectives. In our work, we use “clustering” specifically in the **graph-theoretic sense**, where the goal is to find a single **low-conductance subset** of nodes.
>
> Formally, we aim to minimize the **conductance** of a subset $ S \subset V $, defined appropriately for directed or hypergraphs (e.g., using random walk-based volume and cut measures).
>
> PageRank-based algorithms are then used as heuristics or approximation methods to solve this **local clustering** problem. So while we focus on the development and extension of PageRank-based methods (HyperACL, GeneralACL), the underlying **optimization objective** is clearly defined as minimizing conductance from a seed set rather than clustering in the k-means or hierarchical sense.
>
> We will revise the introduction to explicitly define the clustering objective used in this paper and distinguish it from broader notions of clustering in machine learning and statistics.
>
> > Explanation of “quadratic optimal”
>
> We define “quadratically optimal” in the footnote on page 1, referring to the approximation ratio matching the lower bound up to a quadratic factor in conductance. We agree this is important and will move the explanation into the main text of the introduction for better visibility.
>
> > Theorem 1 & 2
>
> Theorem 1 applies to directed weighted **graphs**, while Theorem 2 extends the result to **EDVW hypergraphs**. The overall proof structure is similar, but the underlying data models and key technical definitions differ. Theorem 2 does not imply Theorem 1 directly, so we present both for clarity and completeness.
>
> > Theorem 1 explanation
>
> We acknowledge that Theorem 1 involves several layered conditions, and it can be nontrivial to parse without going through the full proof. To aid understanding, here is a clarification:
> (1) A random seed set \(S\) determines a corresponding optimal cluster \(S^*\).
> (2) If \(S\) satisfies two mild conditions (which hold with high probability under uniform sampling), then the algorithm returns a quadratically optimal cluster with at least 50% probability over its internal randomness.
>
> So the 50% probability is **not conditional** on \(S^*\), but over the algorithm’s success given that \(S\) meets the conditions. We’ll revise the theorem statement to make this clearer. For example, if we randomly sample 100 starting nodes (each defining a set \(S\)) and all satisfy the two stated conditions, then running our algorithm once on each would yield at least 50 quadratically optimal local clusters, in expectation.
>
>
>
> > On the proof of Theorem 21
>
> We call Theorem 42 with C=S* and D=S. Then, we use Equations (92) and (93) to get either S or S*\S to satisfy the “<=” condition (and this is where the “1/2” probability roots from). The randomness is on D=S. Specifically, (1) in Theorem 1, S is randomly sampled from all subsets of V; (2) in Theorem 42, since $C=S^* \subseteq V$, a random sample of S from V is also a random sample from C, if S happens to be a subset of C.
>
> Because D=S is randomly sampled from V, using the condition of Theorem 21 that S and S*\S share the same optimal local cluster S*. Therefore, the probability of S turning out to be S or S*\S is the same, in other words, Equation 92 and Equation 93 both have 50% chance to hold with a uniformly sampled starting node S from V.
>
> We agree that our current phrasing is misleading and would appreciate further feedback.

---

> > ### Comment · Reviewer_XTcC · 2025-06-07
> > **Followup comments**
> >
> > Thanks for the response, and many of my questions are answered. However, still about the randomness of Theorem 1:
> >
> > I understand that you don’t have to directly condition on S^*, but from your description, it is still about some condition on S, which is random. Then I’m still not sure about your logic about randomness here, because in principle you need to condition on some event of S happens, which means your subsequent analysis cannot use that S is uniformly random because of this condition. Does this create an issue?
> >
> > And from your example about 100 starting nodes, are you suggesting that the 50% probability is not from the randomness of S, but from your algorithm? If so, then why don’t you just say given that S is a fixed set that satisfies the two conditions, your algorithm succeeds with constant probability?
> >
> > Regarding your response about Theorem 21: you said you apply Thm 42 with C = S^* and D = S, but this doesn’t look correct, because in the statement of Thm 42 you need a fixed set C, and then D is a uniform sample of C, and here S^* is dependent on S, no?

---

### Decision · Action_Editor_T45f · 2025-07-02

**Recommendation:** Reject

**Audience:**

No

**Audience Explanation:**

The findings of the paper are in question (see the previous section), so they are therefore of dubious interest to the TMLR audience.

However, if these concerns were to be adequately overcome, I do believe the (revised) findings would be of interest to a segment of the TMLR audience.

**Claims And Evidence:**

No

**Claims Explanation:**

During the initial review phase, the reviewers identified several perceived flaws with the manuscript as submitted:

- concerns regarding the need to compute an NP-hard quantity for the approximation results to hold
- concerns regarding the correctness of the mathematical claims made in the paper
- a perceived gap in the empirical study with respect to measuring running time / scaling
- concerns regarding the clarity of the technical presentation throughout the manuscript

Unfortunately, the author response / discussion phase was not sufficient to adequately mitigate these concerns.

**Resubmission Of Major Revision:**

The authors may consider submitting a major revision at a later time.